# Growth of Neogene Andes linked to changes in plate convergence using high-resolution kinematic models

Felipe Quiero[1], Andrés Tassara [2,3✉], Giampiero Iaffaldano[4] & Osvaldo Rabbia [5]

The Andean cordillera was constructed during compressive tectonic events, whose causes and controls remain unclear. Exploring a possible link to plate convergence has been impeded by the coarse temporal resolution of existing plate kinematic models. Here we show that the Neogene evolution of the Andean margin is primarily related to changes in convergence as observed in new high-resolution plate reconstructions. Building on a compilation of plate finite rotations spanning the last 30 million years and using noise-mitigation techniques, we predict several short-term convergence changes that were unresolved in previous models. These changes are related to main tectono-magmatic events and require forces that are compatible with a range of geodynamic processes. These results allow to revise models of ongoing subduction orogeny at its type locality, emphasizing the role of upper plate deformation in the balance between kinematic energy associated with plate motion and gravitational potential energy stored in orogenic crustal roots.

[1] Universidad de Concepción, Facultad de Ciencias Químicas, Doctorate Program in Geological Sciences, Concepción, Chile. [2] Universidad de Concepción, Facultad de Ciencias Químicas, Departamento Ciencias de la Tierra, Concepción, Chile. [3] Millenium Nucleus CYCLO "The Seismic Cycle along Subduction Zones", Valdivia, Chile. [4] University of Copenhagen, Department of Geosciences and Natural Resource Management, Copenhagen, Denmark. [5] Universidad de Concepción, Instituto de Geología Económica Aplicada, Concepción, Chile. ✉email: andrestassara@udec.cl

The Andean Cordillera, an archetype of subduction-related mountain belts[1,2], is the result of hundred million years of convergence between various oceanic plates and the western margin of South America[3,4]. This process left a marked along-strike segmentation of the continental margin[5,6]. Segmentation is particularly obvious[5,7] when considering variable amounts of Neogene crustal shortening, which ranges from more than 300 km at 20°S to less than 40 km at 40°S. Shortening has traditionally been related to the episodic development of compressional tectonic events that can be broadly recognized in the geological evolution of volcano-sedimentary basins, magmatic arcs and fault systems[3,4,7–10]. However, the geodynamic mechanism explaining the occurrence of these compressional events as well as the causes for the along-strike variability of the associated crustal shortening remain controversial.

This mechanism should consider the dynamic coupling between both converging plates at the subduction zone, connecting compressional tectonic events with changes in plate motion. Such a relationship has been suggested, for instance, to explain an increase in tectono-magmatic activity along with changes in the tectonic style following an acceleration of convergence during the birth of the Nazca plate after breakup of the Farallon plate in the late Oligocene—early Miocene[3,4,9–13]. Available kinematic plate models[14–18] show a subsequent gradual decrease of convergence rate, from nearly 15 cm/yr to half of this value at present. Several authors[19–23] have suggested that this convergence slowdown is mainly related to the growth of the cordillera, as this creates large trench-perpendicular gradients of gravitational potential energy (GPE) that must be laterally supported by high shear stresses at the interplate megathrust, augmenting the frictional resistance to subduction of the Nazca plate.

Quantitative models based on this mechanism point to a widely recognized Mid-Late Miocene tectonic event known as the Quechua phase[4,8–10,24] as the starting point for convergence deceleration[21,23]. This event is related to dramatic changes in shortening rates, magmatism and surface uplift along the entire western margin of South America[3,4,7,10,11,25,26]. However, large uncertainties in Neogene convergence history derived from

available plate kinematic reconstructions have hampered establishment of a clear temporal link between changes in plate convergence and the tectono-magmatic evolution of the continental margin. As a consequence, some authors favor a direct control of upper plate deformation on convergence[7,21–23,27,28] while others observe no relationship between these two processes[11,25,29–32].

Temporal variations of Nazca-South America convergence can be reconstructed by combining finite rotations for the relative spreading motions across mid-oceanic ridges encompassed in the plate circuit Nazca-Pacific-Antarctica-Nubia-South America. Available kinematic reconstructions[14–18] featured a temporal resolution of 5–10 Myr. Recently, there has been progress in mapping Neogene spreading-motion variations at finer temporal resolution[33–37].

In this work we use a compilation of recently published finite plate rotations and statistical tools to mitigate the emergence of noise-related kinematic artifacts that appear at high temporal resolution[38], to obtain a reliable, high-resolution reconstruction of the Nazca-South America convergence during the Neogene.

## Results

**High-resolution plate kinematic model**. In order to obtain stage Euler vectors for the motion of Nazca relative to South America over the past 28 Myr, we combine published finite rotations for the described plate circuit[33–35,37,39]. These data sets mostly feature a temporal resolution of <1–3 Myr and benefit from noise mitigation achieved via the Redback software[38]. We use the covariances associated with each set of finite rotations to map uncertainties on the Nazca-South America stage Euler vectors (see Methods and Supplementary Information).

We use our model and data reported by Somoza and Ghidella (2012; hereafter SG12, ref. [17]) to retro-project the trajectories with respect to South America of points on the Nazca plate that are currently entering the trench axis at 20°S, 30°S and 40°S (Fig. 1). Main features of SG12 are similar to other published reconstructions that encompass the same plate circuit, but SG12 includes uncertainties that can be directly compared with our estimates.

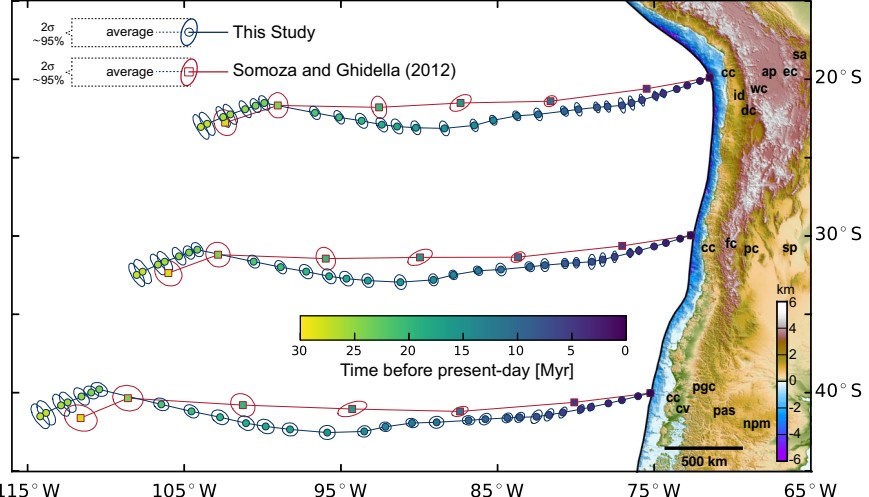

**Fig. 1 Retro-projected trajectories for points of the Nazca plate relative to South America and topography of the Andean margin.** We selected three points on the Nazca plate that are currently entering the trench axis at 20°, 30°, and 40°S, and show their retro-projected position as predicted by our kinematic reconstruction (circles with blue outline) and by Somoza and Ghidella (2012; squares with red outline). Colors for each point represent the average age of the respective stage rotation and the ellipses indicate their 95% confidence interval (i.e. two standard deviations = 2 s). The colored map shows the bathymetry and topography of the Andean margin east of the trench axis. Labels of morpho-structural units that form the anatomy of the margin are also used in Fig. 3 and are defined as follow: cc Coastal Cordillera, id Intermediate Depression, dc Domeyko Cordillera, wc Western Cordillera, ap Altiplano, ec Eastern Cordillera, sa Subandean System, fc Frontal Cordillera, pc Precordillera, sp Sierras Pampeanas, cv Central Valley, pgc Patagonian Cordillera, pas Patagonian System, npm North Patagonian Massif.

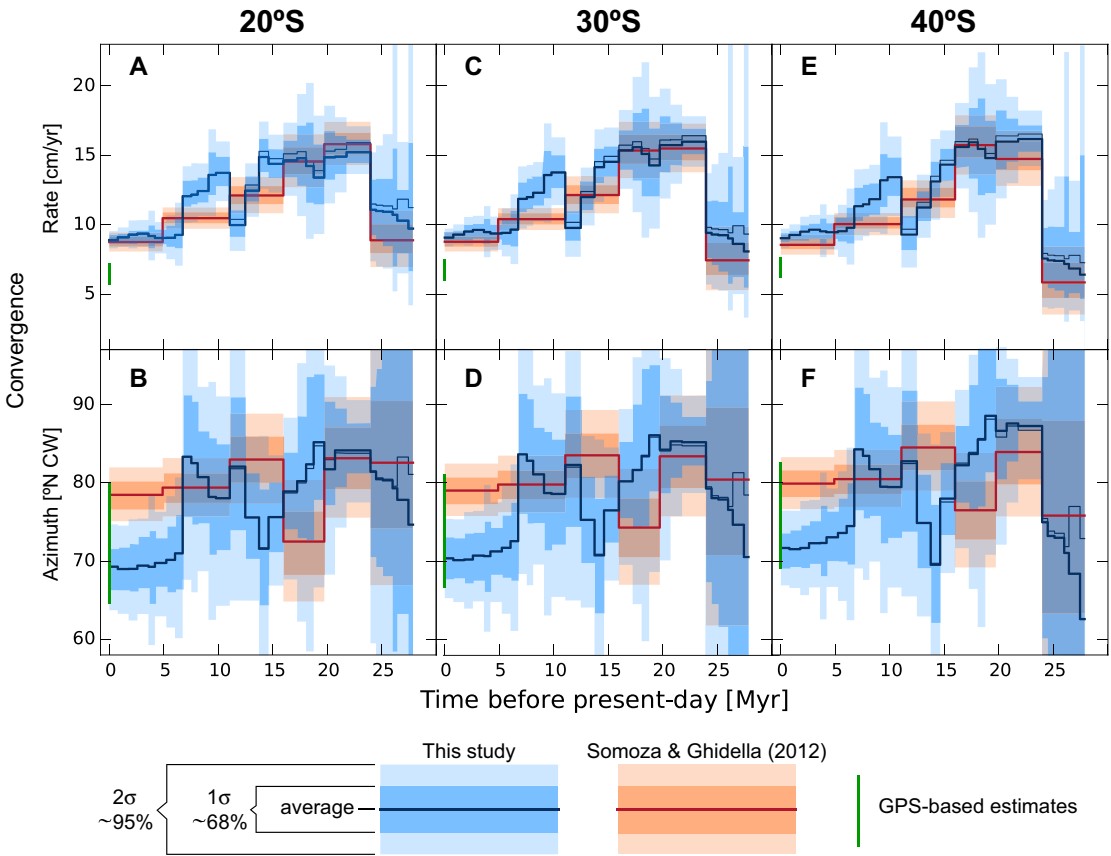

**Fig. 2 Variation of Nazca/South America convergence rate and azimuth at different latitudes along the Andean trench for the last 30 Million years.**
**A** Convergence rate at 20°S from the study of Somoza and Ghidella (2012; ref. [17]) in red and our study in blue. Convergence is inferred from an ensemble of one million samples of the Euler vectors (see Methods) drawn from the nominal values and associated covariances. Solid lines show the ensemble average, while polygons show the one sigma (~68%) and two sigma (~95%) confidence intervals (as shown in the legend of the figure). **B** Azimuth of convergence at 20°S for the same reconstructions shown in **A**. **C, D** Same as **A, B**, but calculated at 30°S. **E, F** Same as **A, B**, but calculated at 40°S. Thin blue line is for an alternative model that includes the relative motion between West and East Antarctica[40,41]. Green lines show the present-day rate and azimuth of convergence derived from various GPS-measured velocities[72-74].

An alternative study[14], which avoids passing through Nubia by using data collected in and around the Weddell Sea, yields kinematics that are in line with those of SG12. The spatial view provided by Fig. 1 is complemented with the temporal variability of convergence rate and azimuth depicted in Fig. 2. From these two figures, we note a first-order coincidence between our results and those of SG12 at the scale of ~5 Myr, like the maximum convergence rate after the birth of the Nazca plate in the Latest Oligocene and its subsequent decrease during the Neogene. However, SG12 imaged this deceleration as a gradual process under relatively fixed azimuth, whereas our model shows some large and rapid (<2 Myr) changes in the rate and azimuth of convergence. Incorporating the Oligocene to Middle Miocene relative motion between East and West Antarctica[40,41] into our reconstruction (Fig. 2) does not alter the temporal pattern of kinematic variations illustrated by our model, nor its main differences with SG12.

At the 68% confidence level (one standard deviation with respect to the average) in Fig. 2, we note statistically significant differences in the uncertainty ranges of SG12 and ours, mostly during the time interval between 15 and 5 Ma, for which differences are also recognized at the more conservative 95% confidence level (two standard deviations). Therefore, we will concentrate our further analyses in the Middle to Late Miocene evolution of convergence. After 20–17 Ma, SG12 shows a steady decreases of convergence rate at intervals of ca. 5 Myr from a peak

of 15 cm/yr to a value of 10 cm/yr at 5 Ma. In contrast, our model predicts that the same magnitude of convergence slowdown occurs in a much shorter period between ca. 15 Ma and 12 Ma. The most relevant feature revealed by our model and the main difference with SG12 is a very rapid convergence acceleration between 11 and 9 Ma reaching a peak of ca. 14 cm/yr all along the entire Andean margin. At 9 Ma our model predicts a decrease of convergence rate to a value around 12 cm/yr that remains relatively constant until 7 Ma when a sudden deceleration to a rate of 9 cm/yr took place. This is accompanied by a notable reduction in the azimuth of convergence from N83°E to N70°E. This value is nearly 10° lower than predicted by SG12, although our convergence rate is very similar to SG12 for the last 5 Myr. Our present-day convergence azimuth lies at the lower bound of GPS-based estimates, whereas our predicted rate is 30–50% larger than current geodetic estimates. This important difference might be related to the megathrust seismic cycle, but this topic is beyond the scope of our present study.

In order to understand the source of differences between our kinematic plate model and SG12 it is important to note that these previous authors utilized a selection of Nazca/Antarctica finite rotations from another study[42] to constrain motions of that part of the Nazca/South America plate circuit. This original study[42] used mapped magnetic lines across the Chile Ridge[43] to infer finite rotations of Nazca relative to Antarctica at chrons 2 A (~3.5 Ma), 3 (~4.9 Ma) and 5 A (~12 Ma). In addition to these,

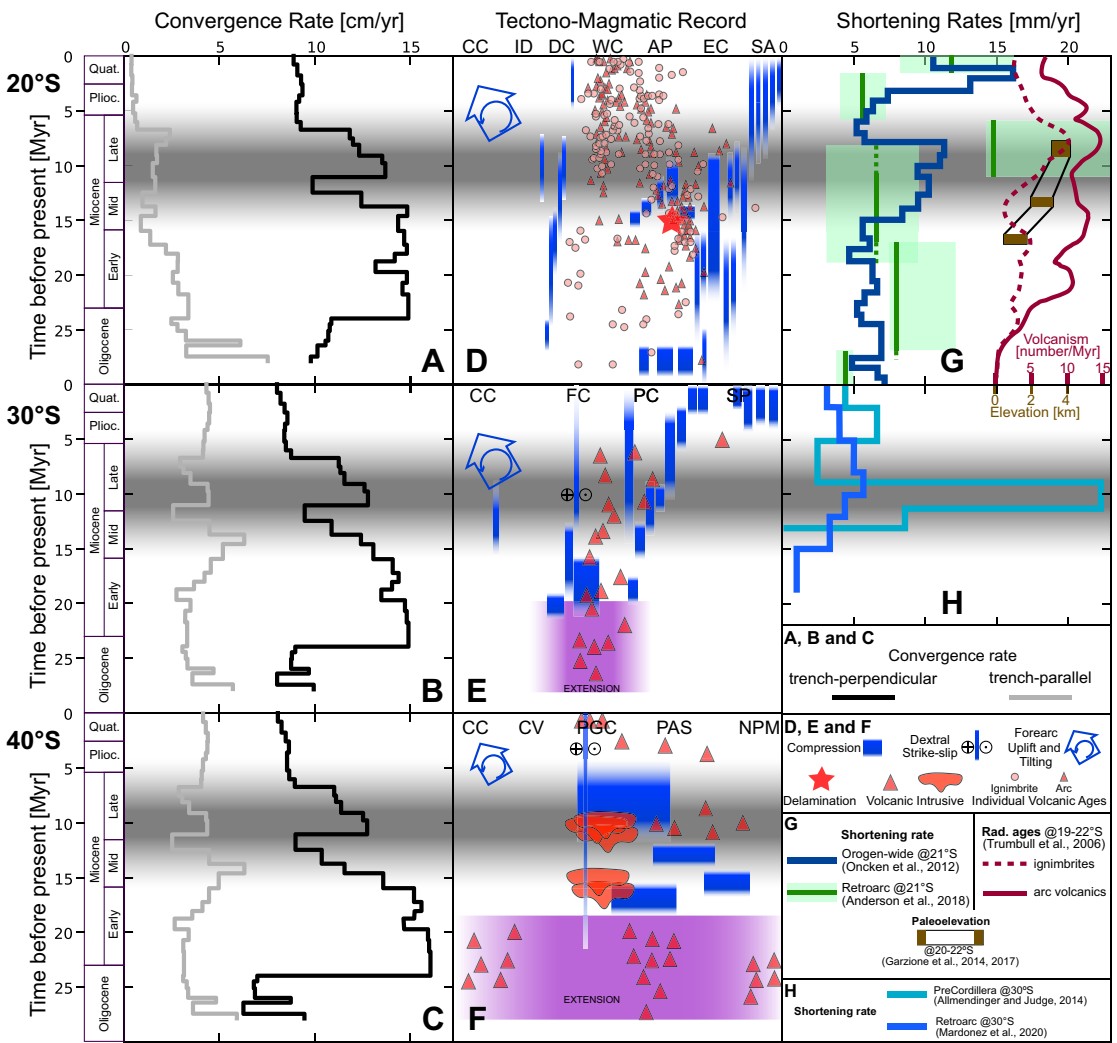

**Fig. 3 Trench-parallel and trench-perpendicular components of convergence velocity compared to the tectono-magmatic evolution of the Andean margin. A–C** shows the temporal evolution of convergence velocity at 20°, 30°, and 40°S (respectively) decomposed on its trench-perpendicular (black line) and trench-parallel (grey line) components (alternatives with time-changing trench axis are available as Figs. S2A–C of Supplementary Information). **D–F** displays a compilation of main tectonic and magmatic events that occurred across the continental margin at 20°, 30°, and 40°S (respectively). Note as a reference the approximate west (left) to east (right) location of morpho-structural units at the top of each panel (labels are defined in Fig. 1). The thickness of blue bars indicates the approximate EW extend of compressive deformation events as informed by original authors (see Supplementary Information). **G**, **H** show estimates of shortening rates at the scale of the whole orogen and for the retroarc region near 20° and 30°S (respectively). **G** also includes time-varying estimates of paleoelevation and variations in the productivity of volcanism (expressed by the number of reported geochronological ages for ignimbrites and arc lavas). Sources of published information used to construct **D** to **H** are described in the Supplementary Information. Horizontal grey bands mark the time interval between 15 and 5 Ma, with emphasis between 11 and 9 Ma, which is the time when our model shows the most important differences with respect to SG12.

the authors (i) interpolated finite rotations at chrons 2 A and 3 to obtain a rotation for ~4 Ma, (ii) interpolated rotations at chrons 3 and 5 A to obtain a rotation at ~11, and (iii) extrapolated chron 5 A to ~16 Ma (see Table 4 in ref. [42]). From these original finite rotations, SG12 utilized the data-constrained rotation at chron 3 (~4.9 Ma), the interpolated one at ~11 Ma, and the extrapolated one at ~16 Ma (see Supplementary Data in SG12). This effectively means that steadiness of Nazca/Antarctica motion is enforced by SG12 between ~16 and ~11 Ma, although the actual data can constrain motion only back to ~12 Ma. We believe this is the main reason for the difference between our reconstruction and SG12 in the period between 15 and 5 Ma.

## Discussion

We compare rates of trench-perpendicular and trench-parallel convergence against the tectono-magmatic evolution of the margin (Fig. 3). These rates were computed for a fixed trench axis (see Supplementary Figs. S2A–C for alternatives considering an evolving trench axis). We focus our analysis on the most relevant and novel features revealed by this comparison, with emphasis in the Mid-Late Miocene evolution associated to the Quechua tectonic phase.

A convergence speedup after the birth of the Nazca plate coincides with the opening of Late Oligocene to Early Miocene extensional basins southward of 28°S[4,13,44,45]. Upper plate extension has been related to an enhanced slab pull and associated rollback as the Nazca slab freely penetrates the upper mantle[45–47]. Basins of this type were absent at 20°S (ref. [9]), perhaps because trench-perpendicular convergence during the Late Oligocene was higher here than southward. Peak convergence rates during the Early Miocene coincides with the beginning of compressive inversion of the Oligo-Miocene extensional basins. This has been

explained as the consequence of the Nazca slab reaching the mantle discontinuity at 660 km depth and getting anchored in the more viscous part of the mantle, which could trigger a stronger coupling with the overriding plate for the rest of the Neogene[45,48,49]. At the Altiplano latitude, this restructuring of the subduction system is associated to a lateral expansion of the area that deforms under a pure-shear mode[50,51] from the center of the plateau to its flanks[31], although shortening rates remain at the moderately high values observed before 20 Ma[29,31].

High trench-perpendicular convergence rates prevailed until 16 Ma in the central and southern transects and until 14 Ma in the north. A rapid convergence deceleration then took place over the Middle Miocene, coinciding with notable changes in the activity of fault systems and magmatic arcs along the entire margin. At 40°S, an eastward migration of the deformation locus seems to be accompanied by a gap of the magmatic arc[52,53], whereas at 30°S the volcanic arc and deformation front start migrating eastward[54,55]. This is related to the beginning of slab shallowing at this latitude[55], a gradual increase of shortening rates across the margin[56] and the initiation of very rapid shortening in the Precordillera fault-thrust belt[57].

At 20°S, the convergence deceleration at 15–14 Ma seems to coincide with the simultaneous activation of fault systems inside and at both flanks of the Altiplano[31,32]. This could be the cause of an increase of orogen-wide shortening rates during the Middle Miocene[31], although rates estimated exclusively in the retroarc remain constant or even lower than previously[29] (Fig. 3). Whatever the case, the Middle Miocene deceleration of convergence was accompanied at the Altiplano latitude by a rapid surface uplift, from a paleoelevation around 1.5 km before 15 Ma to 2.5–3.5 km near 13 Ma[28,58]. This has been linked to delamination of lithospheric mantle and eclogitic lower crust[25,28,30,58,59] that could be responsible for fundamental changes in the thermomechanic conditions under which tectonics and magmatism occur inside the Central Andean plateau.

A sudden acceleration of convergence took place along the entire margin at the beginning of the Late Miocene, with high rates (ca. 14 cm/yr) prevailing during a short period of time between 11 and 9 Ma. This correlates with a westward shift of compressive deformation back to the Patagonian Cordillera and a short phase of renewed magmatism at 40°S[52,53]. At 30°S, an eastward expansion of the volcanic arc is documented[54,55] in synchrony with a rapid propagation of the deformation front inside the Precordillera at extreme shortening rates near 20 mm/yr (ref. [57]). Peaks in orogen-wide shortening rates are documented at this time across 30°S (ref. [56]) and 20°S (ref. [31]), as well in the retroarc region behind the Altiplano[29]. The latter is associated to an eastward shift of deformation into the Subandean fold-thrust belt, dramatically changing the crustal-scale deformation mechanism from distributed pure-shear to localized simple-shear[7,29,31,50,51]. This is followed by an ignimbritic flare-up inside the Altiplano, attesting to a thermally-activated melting of the lower-mid crust[25,60], and the achievement of present-day topography and crustal thickness[28,58].

After the paroxysm of the Quechua phase in the earliest Late Miocene, shortening rates drop at 30°S and 20°S whereas convergence rate seems to decrease in two steps at 9 Ma and then at 7 Ma. This coincides with the installation of hyperarid conditions at the cordilleran western flank[61] as a consequence of the rain shadow effect created by the uplifted cordillera. No erosion means minimal unloading of forearc fault systems that remain largely inactive. According to the simple-shear mode of plateau growth[50,51], the rigid forearc then experienced passive uplift via trench-ward tilting[62] as a response to thickening of the weak lower crust that is pushed by the westward underthrusting of the Brazilian shield below the Subandean system. The eastern side of the orogen concentrates most of the moisture coming from the Amazonas after 5 Ma, meaning enhanced erosional unloading of the fault-thrust belt and concentration of deformation[29,61,63]. An acceleration of shortening rates during the Pliocene at 20°S and 30°S has been related to a climate-driven change toward more humid conditions around 5 Ma[29,31,63]. The associated thickening of weak lower crust is likely redistributed by lateral crustal flow[64,65] and not transferred to the forearc, which could explain why convergence velocity have remained notably stable over the last 5–7 Myrs along the entire margin.

The fundamental reasons explaining the observed correlation between variations in convergence and Andean tectonic evolution remain to be investigated. We attempt a preliminary analysis to elucidate whether the kinematic changes are related to the absolute motion of either plates or both and to highlight the first order features of the forces causing such changes.

We use finite rotations described above to calculate trench motions of Nazca and South America relative to Antarctica, which features a slow absolute motion during the Neogene[66] and thus it arguably resembles an absolute reference frame. Figure 4 demonstrates that most of the changes in convergence between Nazca and South America during the Neogene are due to the absolute motion of the oceanic plate. This is in agreement with global-scale kinematic plate models at low temporal resolution[11] that also show changes in Nazca-South American convergence under a relatively fixed position of the continental plate with respect to the mantle reference frame.

In particular, this indicates that the main difference between our reconstruction and SG12 (i.e. the pattern of convergence deceleration and acceleration coinciding with the Late Miocene Quechua phase) owes ultimately to the Nazca–Antarctica portion of the plate circuit. In this context, there are two aspects worth mentioning in order to lend support to this specific feature of our reconstruction:

i. This pattern of temporal changes of motion is not an artifact arising from the specific temporal resolution at which we interpolate finite rotations and reconstruct motion. In fact, such a pattern remains clear also when we reconstruct the Nazca–Pacific–Antarctica motion at different temporal resolutions (see Methods and Supplementary Figure S3).

ii. There is independent evidence that such changes indeed occurred. In fact, the spreading rate of the Chile Rise between 37°S and 46°S (Fig. 4A) has been directly observed from 17 cross-ridge magnetic lines[43] and exhibits a temporal pattern of changes that is in line with our reconstruction. The agreement remains regardless of the temporal resolution utilized in our reconstruction, particularly when the same resolution than the original data[43] is used (see Methods and Supplementary Figure S3). Furthermore, the short-lived change in Nazca/Antarctica azimuth between ~14 and ~12 Ma that we reconstruct through the Nazca–Pacific–Antarctica circuit is consistent with a ~10° counterclockwise change around 16 Ma in the direction of the Chiloe and Guafo fracture zones originally mapped[43] using ocean-floor observations (Fig. 4B). Lastly, most fracture zones mapped around the Chile Rise from vertical gravity gradients (VGG)[67] also exhibit a similar short-lived change in direction, although the length of the expected differently-oriented portions of fracture zones is close to the limit imposed by spatial resolution and uncertainty of the VGG-mapped fracture zones (see Methods and Supplementary Figure S4).

Accepting that Neogene changes in plate convergence are mostly due to variations in the absolute motion of the Nazca plate, and based on methods and analytical equations derived in

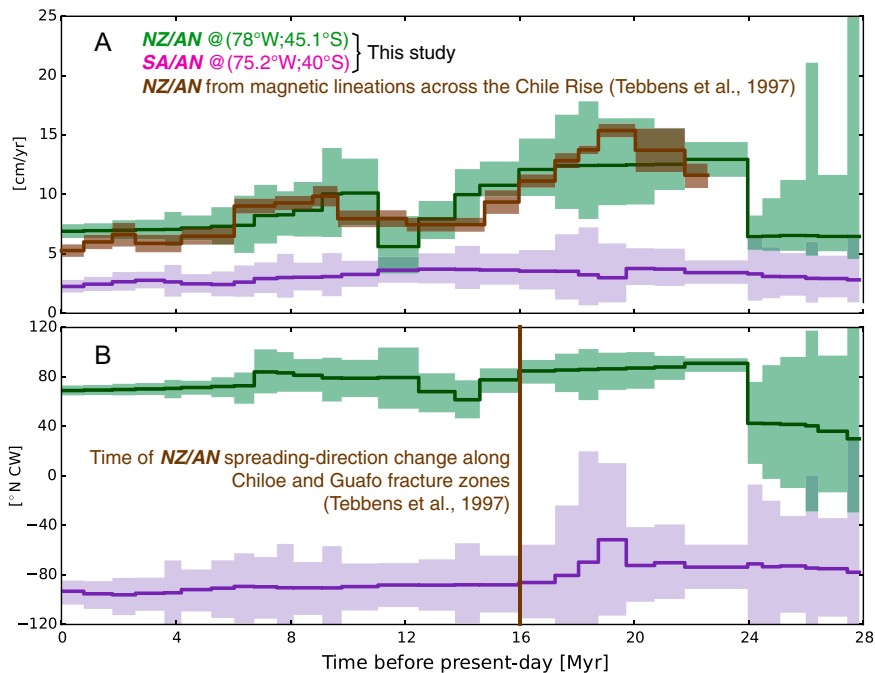

**Fig. 4 Plate motions with respect to Antarctica.** Rate (**A**) and azimuth (**B**) of motions of Nazca (NZ, green) and South American (SA, purple) plates relative to Antarctica (AN) plate, calculated at the specified coordinates. Motions are calculated by sampling one million times the finite rotations discussed in the main text. Uncertainty ranges are reported at the 95% confidence interval. The NZ/AN motion reconstructed here via the Pacific plate is compatible with the independent inference of changes in NZ/AN spreading rates and azimuth derived from observed cross-ridge magnetic lines as reported by Tebbens et al. (1997, in brown).

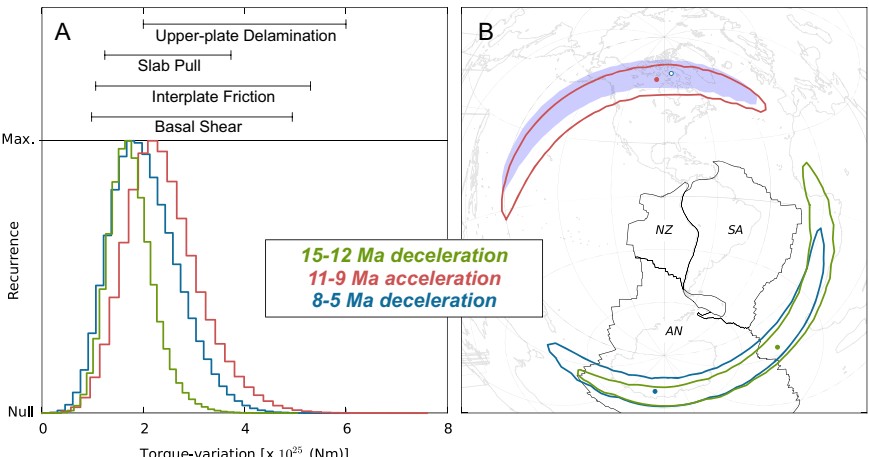

**Fig. 5 Torque variations associates to changes in plate convergence between 15 and 5 Ma. A** Lower panel shows the statistical distributions of torque-variations upon Nazca plate that are needed to generate the Middle Miocene deceleration (between 15 and 12 Ma in green), main Quechua acceleration (between 11 and 9 Ma in red) and Latest Miocene deceleration (between 8 and 5 Ma in blue). Ensembles of torque variations are calculated from one million samples of the Euler-vector variations. Upper panel shows estimates of torque variations associated with possible geodynamic processes acting on Nazca plate (see main text and Methods). **B** Map showing the nominal poles of torque-variation (dots) and the contours where 95% of the samples fall, calculated assuming a viscosity of 5E19 Pa*s in the asthenosphere and of 1.5E21 Pa*s in the upper mantle[71]. The light-blue transparent area is the antipodal region of the blue contour.

previous studies[68,69], we generate an ensemble of one million samples of the torque variation needed to explain the main kinematic changes of Nazca during the Mid-Late Miocene (Fig. 5). This represents an estimate of the magnitude and orientation of the additional torque established upon Nazca that caused its kinematic change, and that accounts for the uncertainties in the reconstructed plate motions. The torque variation during the Latest Miocene deceleration (8–5 Ma) is similar to the changes needed for the Middle Miocene slowdown (15–12 Ma),

and both are virtually opposite in sign and almost equal in magnitude to the variation associated with the main Quechua acceleration (11–9 Ma). This consideration suggests that the kinematic energy gained by Nazca through the syn-Quechua acceleration is later converted (via crustal shortening and thickening) into GPE that is stored in the crustal root, resulting thus in the post-Quechua deceleration.

Simple analytical calculations (see Methods) allow us to infer that the estimated magnitudes of torque variations during and

after the main Quechua acceleration are consistent with—and thus could in principle be caused by—changes of around 5E11 N/m in the slab pull force, or changes near 0.3 MPa in basal shear tractions underneath Nazca. Furthermore, they could also correspond to a change of the friction coefficient of the order of 0.03 occurring uniformly along the entire brittle megathrust, or around 0.1 if this occurred along the interplate fault in front of the central Altiplano-Puna segment. Although simplistic, these analyses indicate moderate changes of dominant forces that could be related to rather rapid, yet plausible variations in the dynamic coupling between the Nazca plate/slab with the upper mantle and/or resistance along its boundary with the upper plate.

In this scenario, changes in the main driving or resisting forces of Nazca would be the main cause of the associated tectono-magmatic events recorded along the continental margin. However, the high temporal resolution of our model allows to observe that the Middle Miocene deceleration does occur shortly after the suggested delamination event at 15–14 Ma (Fig. 3). It can be argued (Methods) that if the gain in GPE caused by delamination was laterally compensated by increasing shear traction at the megathrust, then the torque-variation upon Nazca would be in the range between 2-to-6E25 Nm. This range is indeed compatible with the histograms shown in Fig. 5A, which supports the plausibility of such a mechanism.

Independently of the possible direct effect on convergence deceleration, delamination likely has an indirect but relevant role on the subsequent main Quechua acceleration at the beginning of Late Miocene (11–9 Ma). The post-delamination thermo-mechanical weakening of the mid-lower crust underneath the Altiplano led to a loss of mechanical strength that is related to a notable strain localization towards the Subandean fold-thrust belt[31]. In our view, this lithospheric-scale failure beneath the orogenic axis and strong concentration of shortening toward the foreland could cause the observed convergence acceleration by allowing a rapid eastward advance of the mechanically coupled slab-forearc system with respect to fixed South America. If this hypothesis is correct, it would imply that the thermomechanical modification of upper plate strength at the central segment of the Andean orogen can regulate whole plate convergence, which then has an impact along the entire converging margin as seen in Fig. 3. Finally, and as previously suggested[21–23], the Latest Miocene deceleration (9–5 Ma) can be seen as the kinematic consequence of a further gain of GPE as simple-shear shortening then continues to efficiently accumulate crustal volume along the cordilleran axis.

Our results show that a high-resolution kinematic model for Nazca-South America convergence fosters a more detailed and previously unavailable understanding of fundamental geodynamic process connecting plate motions with the tectono-magmatic evolution of the Andean margin. As more high-resolution reconstructions progressively become available, we anticipate future studies focused on other subduction zones and possibly over longer geological intervals that will help testing hypotheses similar to those explored here. Particularly relevant will be studies of the fundamental role that upper plate deformation along orogenic axes could have on the balance between kinematic energy associated with plate motion and GPE stored in crustal roots.

## Methods

### Reconstruction of the Nazca/South America finite rotations.
We reconstruct the past position of the Nazca plate relative to South America since 28 Ma by combining finite rotations of the past position of Nazca plate relative to the Pacific plate[39], of the Pacific plate relative to the Antarctica plate[33], of the Antarctica plate relative to the Nubia plate[37] and of the Nubia plate relative to the South America plate[34]. The finite rotations of Croon et al. (2008, ref. [33]) feature a temporal resolution of 1–2 Myr and are thus prone to noise impact when calculating stage

Euler vectors. As part of this study, we mitigate the impact of noise in this data set using the Redback software[38] (see Supplementary Information). The finite rotations of DeMets and Merkouriev (2019, ref. [34]) and DeMets et al. (2021, ref. [37]) already benefit from mitigation of the impact of data noise through the Redback software. The finite rotations of Wilder (2003, ref. [39]) feature a temporal resolution of mostly ~3 Myr (the only exceptions being C6Cn.3n and C7n.1n, which are less than 1 Myr apart from each other), and thus are less prone to the impact of noise. We utilize them in their original form, after excluding the rotation associated with anomaly C5Dn (as this is the least constrained of the dataset and features covariances up to 10 times larger than any other rotation in the dataset) as well as C6Cn.3n (in order to avoid potential noise issues arising from the temporal proximity to the relatively-less-uncertain C7n.1n rotation). We re-assign ages to all finite rotations according to the GTS12 geomagnetic reversals timescale[70]. For each data set, we sample one million times each finite rotation using the nominal value and the associated covariance matrix. This allows us to generate an ensemble of one million realizations, or samples, of the finite rotation temporal set. We use each sampled set to perform an interpolation of finite rotations at the times steps younger than 29 Ma of the reconstruction of DeMets and Merkouriev (2019, ref. [34]), which adopts the astronomically-tuned GTS12 timescale. This allows us to obtain an ensemble of one million samples of each finite rotation data set that feature the same (interpolated) ages. We calculate the average (i.e. nominal) values and the associated covariances for each of the interpolated finite-rotation data sets and report them in the Supplementary Information. Next, we combine these data sets in order to obtain an ensemble of one million samples of the finite rotations that reconstruct the past position of the Nazca plate relative to the South America plate since ~28 Ma. From these, we calculate the average (i.e., nominal) values and the associated covariances, which we report in the Supplementary Information. From such an ensemble, we calculate an ensemble of one million samples of the Stage Euler vectors for the Nazca/South America past relative motion. We report in the Supplementary Information the average (i.e., nominal) values and the associated covariances for each temporal stage. In an alternative reconstruction, we repeat the same procedure described above, but this time we include the relative motion between East and West Antarctica between ~30 and ~11 Ma, reconstructed from the studies of Granot et al. (2013 ref. [40], and 2018 ref. [41]). Finite rotations and stage Euler vectors for this alternative reconstruction are also provided in the Supplementary Information.

### Focusing on the Nazca/Antarctica motion.
From the reconstruction illustrated in Fig. 4, we utilize the portion Nazca–Pacific–Antarctica to isolate the motion of Nazca relative to Antarctica in order to highlight that the pattern of convergence deceleration and acceleration around the Quechua phase owes to changes in the absolute motion of Nazca. This also allows us to make an explicit comparison between our reconstruction and the Nazca/Antarctica spreading history put forth by Tebbens et al. (1997, ref. [43]) on the basis of observed magnetic lines across the Chile Rise. Following the same procedure illustrated above, we repeat the sampling of Nazca/Antarctica motion using three different sets of ages, in addition to the one of the reconstruction by DeMets & Merkouriev (2019, ref. [34]) used in this study. These are (i) the ages of Wilder (2003), (ii) those of Croon et al. (2008), and (iii) those of Tebbens et al. (1997). Results can be inspected in Supplementary Figure S3. In all three cases, the kinematic pattern around the Quechua phase remains visible. In the case featuring the resolution of Tebbens et al. (1997) our reconstruction is in good agreement with the spreading history inferred from magnetic lines across the Chile Rise. The short-lived change of azimuth between ~14 and ~12 Ma (see Supplementary Figure S3b) implies that ~70-km-long segments of the ocean-floor fracture zones (FZs) should be differently oriented (~10° counter-clockwise) relative to the rest of the FZs tracks. This is in fact the case for most FZs identified around the Chile Rise from Vertical Gravity Gradients[67], as illustrated in Supplementary Figure S4 where we compare the direction of FZs from points identified on ocean-floor whose age is between ~14 and ~12 Ma with the same-FZs tracks before and after that period. The change is not always evident on both sides of the identified FZs, but it is on at least one side of any FZs. This may owe to the uncertainty associated with the identified FZs since Matthews et al. (2011) state an uncertainty on the points along FZs of ~6 km, and in some cases up to 10 km. This means that the 14–12 Ma azimuth (red dashed lines in Supplementary Figure S4) identified from FZs points that are ~70 km distant and uncertain by ~6 km comes with an uncertainty of ±5°, which is comparable with the azimuth change that we aim at resolving.

### Estimate of torque-variations required for plate-motion changes.
Previous studies[68,69] provided equations to calculate the torque-variation necessary upon a tectonic plate in order to generate a reconstructed kinematic variation. Specifically, by differentiating the torque-balance equation at two distinct moments in geological time, one obtains a relationship where the torque-variation vector is linked to the Euler-vector change through a linear map that depends on (i) the shape of the tectonic plate, and (ii) the ratio between viscosity and thickness of the underlying asthenosphere (see equations in[68]). Here we utilize the shape-files of Nazca put forth by the Earthbyte initiative (www.earthbyte.org) at the stages indicated in Fig. 5. Furthermore, we assume an average viscosity of the asthenosphere of 5E19 Pa*s, in line with previous inferences derived from modelling of glacial isostatic adjustment data[71]. The thickness of the asthenosphere (here set to 150 km) is

determined from the relationship linking it to the asthenosphere viscosity contrast relative to the upper mantle[71], whose viscosity here is assumed to be the Haskell-like value of 1.5e21 Pa*s. Such a relationship is inferred from the fit of glacial rebound models to long-wavelength rebound data. We utilize the equations above in order to convert ensembles of one million samples of the Euler-vector change drawn from our reconstruction of Nazca–Antarctica relative motion (Fig. 4) into ensembles of torque-variations, the distributions of which are shown in Fig. 5.

**Simple analytical estimates of dominant controls associated with torque-variations.** Generally speaking, the dominant controls on plate-motion changes can be either plate-boundary forces or shear stresses at the base of the plate. Therefore, in order to make a first-order estimate of force variations, one shall divide the torque-variation magnitude estimated from our kinematic reconstruction by the product of Earth's radius with (i) the length of the tectonic margin where plate-boundary forces are assumed to change, or alternatively (ii) the basal area of the plate. The Nazca convergent margin between 15 and 5 Ma featured a length of around 6.5E6 m, while the Nazca basal area was around 1.55E13 m². The majority of the Mid-Late Miocene torque-variation samples are in the range 1E25 to 4E25 Nm (Fig. 5A lower panel). If these were caused by changes in forces acting along the Andean convergent margin, then such changes must have been in range from 3E11 to 9E11 N/m. These could be directly associated to changes in the net slab-pull force and/or to changes in the frictional resistance along the brittle margin. In the latter case, the force-variation would require a change in the coefficient of friction in the range from 0.01 to 0.05 along the entire length of the brittle interface (assumed to extend to a depth of around 30 km). The estimated change in friction coefficient scales linearly with the length of the margin where friction varies. Therefore, assuming that the coefficient of friction changed only along the sediment-starved Central Andean margin this would yield an estimate for the change in friction in the range between 0.03 and 0.15. Instead, if the torque-variations were caused by changes in Nazca basal shear stresses, then such changes must have been in the range from 0.1 to 0.5 MPa. Lastly, if one assumes that delamination of the thick and dense Andean crust and upper mantle generates uplift, then the GPE gain associated with such uplift must compensate for the change in buoyancy caused by the removal of dense eclogitic crust and lithospheric mantle material. Thus, the force exerted upon Nazca would be at most equal to the change in buoyancy associated with the delamination process. The buoyancy change depends on the volume of delaminated material and its density contrast with the mantle wedge. An estimate[59] of the cross-sectional area of removed material at 20°S is of the order of 30.000 km², resulting from a delaminated region that is 80 km thick and 350 km width. A summary of previously published seismic models[28] allows to infer a minimum along-strike extend of 200 km for a region between 19° and 21°S with low velocities at lithospheric mantle depths that can be related to delamination. This means a volume of 6E15 m³ of delaminated material, which multiplied by a positive density contrast with the mantle wedge in the range from 100 to 150 kg/m³ (as for example in ref. [59]) then implies a torque variation in the range from 2 to 6E25 Nm.

## Data availability
All data used in this study are referenced and compiled as a database that is available in the Supplementary Information as Source Data Files.

## Code availability
Codes used in this study are referenced.

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

## Acknowledgements

This research was supported by the Chilean ANID Milenio Grant NCN19_167 Nucleo Milenio CYCLO (A.T.), the Geology Section at the Department of Geosciences and Natural Resource Management of the University of Copenhagen (F.Q. and G.I.), and BHP-Metals Exploration Chile (F.Q. and O.R.).

## Author contributions

G.I., O.R., and A.T. conceived and developed the main ideas and designed the study. F.Q. and G.I. compiled the finite rotations database and performed the plate kinematic reconstruction. G.I. computed torque-balance calculations. G.I. and A.T. calculated forces. A.T. created and edited main figures and wrote the main text. All authors collaborated in writing and revising the manuscript.

## Competing interests

The authors declare no competing interests.
