## [Peer Review File · Nature Communications]

REVIEWER COMMENTS

Reviewer #1 (Remarks to the Author):

The manuscript presents a two-part test of the idea that variations in the rate of Andean uplift and plate convergence rate are a response to seesaw-like changes in the balance of the forces related to plate convergence and gravitational stress from the orogen, exerted across the subduction interface.

The first part of the test involves constructing a high-resolution model of relative Nazca-South American plate motion by the addition of previously-published high-resolution and noise-reduced rotations in the circuit Nazca-Pacific-Antarctica-Nubia-South America. The temporal resolution achieved comfortably exceeds that of any previous construction of any circuit for NAZ-SAM motions. By comparing convergence rates and azimuths to the geological record of Andean convergence at three points along the trench, the authors conclude there to be a clear temporal correlation between periods of fast convergence and periods of uplift.

The second part of the test involves constructing a model of the plate-driving torque balance that could have accompanied one of the phases of acceleration and subsequent deceleration. The authors interpret the distribution of these torques to be consistent with the idea that plate convergence rate increases can transmit mountain-building stress to the upper plate via the subduction interface, and that gravitational energy consequently stored in the orogen can be the cause of a subsequent deceleration in convergence rate.

The methods used are at the forefront of the field, the conclusions reached are appealingly straightforward and plausible. The work is presented in clear, well-organized prose, and illustrated for the most part with uncluttered and easily-read figures that are well incorporated with the text.

Despite this, I have a number of concerns about the plate kinematic modelling that I think need to be responded to before considering this work further, either here or for another journal:

1. Statistical comparison of the new model and previous models. The model by Somoza and Ghidella is rightly chosen for the statistical comparison, because of the availability of suitable covariances for calculation of confidence regions. The uncertainties of the two methods are discussed and compared at 68% (1-sigma) level, which I note to have been done before for studies using Redback for noise reduction, but which I also note is less conservative than the 95% (2-sigma) uncertainties used in the majority of other studies in the field (including Somoza and Ghidella's). For this journal, and for reasons that I will come to, I would find the conclusions of the first test to be more convincing if the discussed differences between the two models could be proved at 2-sigma, rather than 1-sigma, level.

2. Inferred vs. constrained rotations in the new construction. Looking at the tables for the various sets of rotations for plate pairs that have been added, I think that the high (2-3 Myr) resolution of constrained rotations (that is, a NAZ-SAM rotation that is constrained by a magnetic anomaly fitting exercise on *every* leg of the circuit used to calculate it) is interrupted over the period C5D-C5 (17.5-11.05 Ma), during which neither the ANT-AFR or AFR-SAM models of DeMets and others features any rotations. The 1-Myr stepped rotations and the changing speeds and azimuths calculated from them and displayed in Figs. 1 and 2 over this period must thus be at least partially inferred. This is important for the rest of the study, because the 17-11 Ma period envelopes and so defines one complete slow-fast-slow cycle with no synchronous named pan-Andean compressive event of its own, and in whose absence there may be no sharp slowdown to correlate with the end of the Pehuencha phase, and no sharp speed up to correlate with the start of the Quechua. In this case, the new model's results would more closely resemble those of Somoza and Ghidella, by showing a long-period slowdown in convergence rates at all studied latitudes. I am suspicious of the model in the 17-11 Ma period not only because of the absence of any pan-Andean tectonic signal, but also because of the unusually short-lived and marked ($\sim 40^\circ$) swerves in calculated NAZ-SAM azimuths. The severity of these swerves, given the stability of SAM-ANT motions and relative fixity of ANT over the same period (as noted in the manuscript's Figure 4) ought to be attributable to changes in the absolute motion of the Nazca plate, and yet appears not be reflected in the orientations of appropriately-aged segments of

fracture zones formed at the flanks of the divergent NAZ-ANT plate boundary, the Chile Rise.

3. Range of previous models. Alongside Somoza and Ghidella's, the study cites three additional previous plate kinematic works on NAZ-SAM convergence. Together, the four cited studies represent a picture of the evolution in the accuracy and resolution of models that have been available to be combined in the Nazca-Pacific-Antarctic-Africa-South America plate circuit. The new set of rotations is, in this sense, a continuation of that evolution. There is however at least one other way around the circuit that has been used, but that is not cited. This is to skip the ANT-AFR-SAM leg in favour of an ANT-SAM model derived from data in and around the Weddell Sea (<http://dx.doi.org/10.1016/j.gloplacha.2014.08.002>). Whilst the model built in this way is neither of particularly high temporal resolution nor completed with useful covariances, what it does offer is the potential to highlight consistent, and therefore robust, features in the convergence history that are at least partially independent of the detailed geometry of the circuits used. Importantly, here, and referring back to my point (2), above, this alternative construction of the circuit yields convergence histories that more similar to that from Somoza and Ghidella's model than to this manuscript's one. These similarities are for a general and sustained decrease in NAZ-SAM convergence rates, and relatively unperturbed azimuths (consistent with the observable sedate changes in fracture zone azimuths on the flanks of the Chile Rise), following the onset of the Pehuenche phase around 25 Ma. With this observation, I think it becomes more important for you to (i) be able to distinguish the new results from Somoza and Ghidella's with increased statistical rigour and, more importantly (ii) explain why the large change in NAZ-SAM azimuth at 17-11 Ma is not reflected in the shapes of Chile Rise fracture zones of a similar age, and (iii) explain why the speed-up at 15 Ma is not reflected in a whole-Andean compressive pulse, contrary to the general conclusions of your study, but seemingly at best only a local uplift near 20°S.

Minor observations:

- Abstract - I think the nature of compressive pulses is clear, but the nature of their causes and controls is not.

- Line 23 - "primary" should be 'primarily'

- Inclusion of West-East Antarctic relative motion in the circuit. Nobody has ever done this, and yet the motion is real and may have lasted until as late as 11 Ma (<https://www.nature.com/articles/s41467-018-05270-w>).

- Fig. 3 - here, I think less really could be more. I find the figure quite difficult to use due to the weight of information that is not used in the text, or material that opens more questions than it answers. For example, was there really no intrusive activity at 30°S or 20°S since 30 Ma, despite abundant evidence for volcanism? As well as this, the thicknesses of the blue bars in parts D, E, and F vary over a greater range than in the key. What is the significance of these changing thicknesses? Which of the contrasting shortening estimates shown in G and H should readers be paying most attention to? Why do neither of them show any increase in shortening for the Pehuenche? In this last respect, I would have appreciated the introductory text being used to more clearly and rigorously define the reality and synchronicity of these phases along the Andes using up-to-date techniques and data.

- paragraph at lines 157-166: Firstly, if you do intend to keep a figure with the level of detail currently included in Figure 3, why not include these considerations of the altiplano processes there instead? Secondly, if the Juan Fernandez Ridge trajectory is not yet calculated at the resolution of the new model, why show it in Figure 3 at all?

- line 199 . "fussing" should be 'fusing'.

Finally, whilst I recognize that my comments are likely to mean a considerable amount of work, either for you to rebut them or to revise your models and the manuscript, I would like to reiterate that I very much admire the rigorous approach you have taken and the high quality of the description of your work. I hope that you will be able to take and use my comments in the constructive spirit that I compiled them.

Kind regards,

Graeme Eagles

Reviewer #2 (Remarks to the Author):

The present manuscript presents an improvement in the temporal resolution of the convergence rates during the Neogene between the Nazca and South American plates. By using this improved temporal resolution, authors attempted to correlate changes in convergence rates with tectono-magmatic phases recognized along three Andean transects, which allowed them to hypothesize about the processes responsible for this correlation. However, the concept of synchronic and discrete tectonic phases in the Andes has fallen in disuse, the temporal resolution behind geological events is not comparable with the higher resolution obtained in this work and, more importantly, correlation does not mean causal relation. Although the accumulation of gravitational potential energy as a trigger for contraction is intriguing, its conversion from kinematic energy due to convergence acceleration is not supported by available data, and more work should be done in order to better understand this process. In the following paragraphs I will address each of these main concerns and some other minor issues I have detected in the present manuscript.

Concept of tectonic phases

When referring to the Pehuenche, Quecha and Diaguita tectonic phases, authors cite works from Mpodozis and Ramos (1989), Charrier et al. (2007), Maloney et al. (2013) and Horton (2018). However, this is a proposal made by Steinmann (1929) designed for a narrow segment, the Western Cordillera of Central Peru, that was never widely adopted for the entire Andes. It was not until the 1970s that a single group of French workers resurrected the terminology and claimed it was Andean wide. However, nothing but contradictions have arisen when geologists tried to apply this concept throughout the Andes, making the notion of Andean building through discrete and synchronic pulses not sustainable. Of the four works cited by the authors in this manuscript, only Charrier et al. (2007) mentions one of these phases, the Quechua, but most recent works, including the compilation made by Horton (2018), do not make use of any of these terms because they are obsolete. Therefore, most readers will not be familiar with these phases and their timing, and the lack of any kind of background given by the authors makes it impossible to assess whether the results are meaningful. In the end, given the uniqueness of each sector of the Andes, this kind of simplifications and global explanations behind shortening phases always fall short.

Correlation between convergence changes and tectonic regimes

Authors propose that after an acceleration in convergence rates, the kinematic energy transmitted from the lower plate will be stored as GPE in the roots of the mountain belt, which will act as a trigger for shortening phases. However, several parametric studies, geodynamic models and regional geological compilations of past and present deformational stages have shown that it is the absolute motion of the overriding plate the main responsible behind the tectonic regime along the margin (Heuret and Lallemand, 2005; Lallemand et al., 2005; Sobolev and Babeyko, 2005; Schellart, 2008; van Dinther et al., 2010; Ramos, 2010; Horton, 2018, among many others). Convergence speed-ups have been shown to be correlated with opening of extensional basins between 27°-46°30'S, and not with their closure (Fennell et al., 2018 and references therein). Moreover, the Salar de Atacama basin at 23°S is an extensional basin developed during the late Oligocene-early Miocene convergence acceleration (Pananont et al., 2004; Jordan et al., 2007; Rubilar et al., 2018), making the claim that higher rates inhibited the opening of extensional basins in the northern transect untrue. On the other hand, it has been shown that the deceleration of the convergence is correlated with the beginning of the Miocene contractional phase, which is produced by the difficulty of the oceanic plate when penetrating the lower mantle given the existing viscosity jump (Quinteros and Sobolev, 2013).

This has led to recent proposals of slab anchoring as the trigger behind the switch from extension to compression observed along the Andes (Chen et al., 2019).

Correlation between convergence changes and magmatic loci

Authors claim that convergence speed-ups correlate with eastward propagation of the magmatic arc. However, it has already been demonstrated through a great deal of works that an acceleration of convergence enhances corner flow in the asthenospheric wedge, favoring the production of magmatism at lower depths over the slab and, therefore, closer to the trench (England et al., 2004; Syracuse and Abers, 2006; England and Katz, 2010; Maunder et al., 2019; among many others). This is exactly the case during the late Oligocene – early Miocene speed-up, evidenced by the presence of widespread magmatism in the present forearc area (see Fennell et al., 2018 and references therein). Once the convergence decelerates, magmatism is produced at higher depths and, in consequence, the magmatic arc migrates eastwards. Therefore, the eastward migration of magmatism cannot be caused by convergence speed-ups, proving that correlation does not mean causal relation.

Andean Tectono-magmatic evolution along selected transects

The compilation presented by authors is incomplete and out of date, and the most recent data supported by improved geochronological, thermochronological, sedimentological and structural studies do not correlate with the convergence accelerations obtained in the model. At 20°S, Anderson et al. (2018) show that the highest shortening rates fall between 11 and 6 Ma, in contrast with Oncken et al. (2006)'s rates. The paleoelevation results cited by the authors are under debate, and others lines of investigation point towards a slow and steady rise of the Andes instead of growth by pulses (see Ehlers and Poulsen, 2009 and Poulsen et al., 2010), which go against the correlations and hypothesis of the authors. At 30°S, Jones et al. (2016) show ongoing extension until 20 Ma, which coincides with the convergence speed-up detected in this manuscript and in previous works. This extensional stage has been recently confirmed by González et al. (2020), who interpret the end of this event occurring in the early Miocene, when convergence rates decelerated. Shortening rates at 30°S recently published by Mardonez et al. (2020) differ from the ones estimated by Allmendinger and Judge (2014), showing that the highest rates occurred between 8 and 5 Ma, uncorrelated with any of the phases proposed in the manuscript. It is surprising that the authors do not compile any information at 40°S, given the wealth of published articles in this area (see Echaurren et al., 2019 and references therein). For example, Ramos et al. (2015) and Folguera et al. (2018) propose several shortening events during the Neogene, but none of them coincide with the acceleration in convergence rates detected in this work. Moreover, extensional deformation at these latitudes overlaps with high convergence rates (see Fernandez Paz et al., 2019), in contrast with the hypothesis developed in this manuscript. In conclusion, if authors made a more exhaustive and complete compilation of the Andean tectonic phases at the selected latitudes, they would find that there are no correlations, that the timing between both datasets is not comparable, or that a lag time might exist between speed ups and shortening. The only existing correlation supported by actual data is a switch between a brief and widespread extensional event towards a shortening phase that occurred in the early Miocene, although this is not something synchronic along the margin. The existence of "sub-phases" within the Miocene contractional stage is not supported by any of the most recent datasets supported by modern geo- and thermochronological analyses.

Minor issues

Although I agree that the 50% difference between present convergence rates and GPS measurements detected by authors could be attributed to the megathrust cycle, I encourage authors to elaborate on this subject and run tests of their model in other settings devoid of this kind of cycles in order to test its validity.

Given that it is the absolute motion of the overriding plate the first order parameter controlling the tectonic regime, taking the South American plate as an almost absolute reference frame seems odd. Even small differences (extension switched to compression after the opening of the Atlantic Ocean at low rates) might affect the overall tectonic regime, something worth investigating.

The idea of delamination occurring in the northern transect affecting the whole Andean margin is speculative and unsupported, and authors should elaborate more before making this proposal. What happened to the north of 20°S? Would this process be synchronic or evolve gradually? Are there any models or works that support this idea?

I will be glad to clarify any of my comments and share any of the papers referenced in the revision and attached PDF.

Sincerely,

Dr. Lucas Fennell

References to papers mentioned in the revision not included in main manuscript:

- Heuret and Lallemand, 2005. Plate motions, slab dynamics and back-arc deformation.
- Lallemand et al., 2005. On the relationships between slab dip, back-arc stress, upper plate absolute motion, and crustal nature in subduction zones.
- Sobolev and Babeyko, 2005. What drives orogeny in the Andes?
- Schellart, 2008. Subduction zone trench migration: slab driven or overriding-plate-driven?
- van Dinther et al., 2010. Role of the overriding plate in the subduction process: Insights from numerical models.
- Ramos, 2010. The tectonic regime along the Andes: Present-day and Mesozoic regimes.
- Fennell et al., 2018. The role of the slab pull force in the late Oligocene to early Miocene extension in the Southern Central Andes (27-46 S): Insights from numerical modeling.
- Pananont et al., 2004. Cenozoic evolution of the northwestern Salar de Atacama Basin, northern Chile.
- Jordan et al., 2007. Cenozoic subsurface stratigraphy and structure of the Salar de Atacama Basin, northern Chile.
- Rubilar et al., 2018. Structure of the Cordillera de la Sal: A key tectonic element for the Oligocene-Neogene evolution of the Salar de Atacama basin, Central Andes, northern Chile.
- Quinteros and Sobolev, 2013. Why has the Nazca plate slowed since the Neogene?
- Chen et al., 2019. Southward propagation of Nazca subduction along the Andes.
- England et al., 2004. Systematic variation in the depths of slabs beneath arc volcanoes.
- Syracuse and Abers, 2006. Global compilation of variations in slab depth beneath arc volcanoes and implications.
- England and Katz, 2010; Melting above the anhydrous solidus controls the location of volcanic arcs.
- Maunder et al., 2019. Modeling slab temperature: A reevaluation of the thermal parameter.
- Ehlers and Poulsen, 2009. Influence of Andean uplift on climate and paleoaltimetry estimates.
- Poulsen et al., 2010. Onset of convective rainfall during gradual late Miocene rise of the central Andes.
- Jones et al., 2016. The role of changing geodynamics in the progressive contamination of Late Cretaceous to Late Miocene arc magmas in the southern Central Andes.
- González et al., 2020. The Iglesia basin in the southern Central Andes: A record of backarc extension before wedge-top deposition in a foreland basin.
- Mardonez et al., 2020. The Jáchal river cross-section revisited (Andes of Argentina, 30° S): Constraints from the chronology and geometry of neogene synorogenic deposits.
- Echaurren et al., 2019. Tectonic controls on the building of the North Patagonian fold-thrust belt (~ 43° S).
- Ramos et al., 2015. The North Patagonian orogenic front and related foreland evolution during the Miocene, analyzed from synorogenic sedimentation and U/Pb dating (~ 42° S).
- Folguera et al., 2018. Constraints on the Neogene growth of the central Patagonian Andes at the latitude of the Chile triple junction (45–47 S) using U/Pb geochronology in synorogenic strata.
- Fernandez Paz et al., 2019. Constraints on trenchward arc migration and backarc magmatism in the north patagonian andes in the context of nazca plate rollback.

Reviewer #3 (Remarks to the Author):

This is a very good paper, addressing a fundamental aspect of crustal evolution, namely the growth of Andean-style mountain belts and possible links to global plate motion, especially convergence velocity. The more detailed kinematic plate motion model presented by the authors is an important step in hypothesis testing. Some minor wording changes would help to clarify the arguments (see attached document) and I have a few more substantive comments on figures that could also help to clarify their arguments (below).

The Introduction nicely sets the stage for the paper, reviewing past work including papers that have explicitly linked slowing convergence to shortening via mountain growth and augmented frictional resistance on the megathrust. In this regard the 1999 paper by Norabuena (GRL 26) was I believe one of the first and perhaps should be referenced.

Figure 1 shows retro-projected points, but despite a very long caption, never says what the points represent. Are these points evenly spaced in time (e.g, 1 million years) so that speed variations can be inferred from point spacing? If not, what do the points represent? There are also some labels on South America, but they are difficult to read against the color background. The geologic details shown are not necessarily important to the arguments presented. I suggest the authors revise this figure to focus on things that are important to their argument.

Figure 3 is similarly very "busy" and could be improved. The convergence rate plots (left hand panels) show a high rate and a low rate line, which presumably represent trench-parallel and trench-perpendicular, but on my print out both come out as dark lines. I suggest making the trench-parallel one either much lighter, or dashed. Alternately consider omitting it - is it necessary?

Related to Figure 3, it seems to me a key point of the paper is a correlation between convergence rate between the plate and shortening rate in the upper plate. Perhaps a scatter plot comparing these two quantities would be useful to explicitly show the connection.

REBUTTAL LETTER FOR REVIEW OF THE MANUSCRIPT:

Growth of the Neogene Andes linked to changes in plate convergence using high-resolution kinematic models

We greatly appreciate the thorough reading of our manuscript done by each reviewer and their intention to contribute for the improvement of the quality and clarity of this work. We have carefully considered all their comments, criticisms and suggestions, as can be appreciated throughout this rebuttal letter and the attached documents. Particularly, the document Changes_Noted include all the modifications done to the original manuscript, which finally derived in the new version of our paper.

Below we tried to isolate specific comments/suggestions of each reviewer and in order to give an answer (in blue) to all of them. For most of these answers, we identified the line numbers in the document Changes_Noted where the implied modifications were included.

Some of the comments and criticisms expressed by Reviewers 1 and 2 implied that we had to regenerate the entire plate kinematic model and modify the interpretative scheme used to link changes in plate convergence with tectono-magmatic evolution of the Andean margin. We remark that main features of the model remain and stand much more robustly than in the previous version, differences with previous models are also clearly delimited for the time interval 15-5 Ma, which coincide with the widely-recognized Quechua tectonic phase that is now the main focus of the paper. These changes impacted in modifications in the title, abstract, main text and figures.

We think that all these changes really help improving the quality of our work and its potential impact. We hope that the Editor and reviewers will agree with us in this respect.

REVIEWER COMMENTS

Reviewer #1 (Remarks to the Author):

The manuscript presents a two-part test of the idea that variations in the rate of Andean uplift and plate convergence rate are a response to seesaw-like changes in the balance of the forces related to plate convergence and gravitational stress from the orogen, exerted across the subduction interface.

The first part of the test involves constructing a high-resolution model of relative Nazca-South American plate motion by the addition of previously-published high-resolution and noise-reduced rotations in the circuit Nazca-Pacific-Antarctica-Nubia-South America. The temporal resolution achieved comfortably exceeds that of any previous construction of any circuit for NAZ-SAM motions. By comparing convergence rates and azimuths to the geological record of Andean convergence

at three points along the trench, the authors conclude there to be a clear temporal correlation between periods of fast convergence and periods of uplift.

The second part of the test involves constructing a model of the plate-driving torque balance that could have accompanied one of the phases of acceleration and subsequent deceleration. The authors interpret the distribution of these torques to be consistent with the idea that plate convergence rate increases can transmit mountain-building stress to the upper plate via the subduction interface, and that gravitational energy consequently stored in the orogen can be the cause of a subsequent deceleration in convergence rate.

The methods used are at the forefront of the field, the conclusions reached are appealingly straightforward and plausible. The work is presented in clear, well-organized prose, and illustrated for the most part with uncluttered and easily-read figures that are well incorporated with the text.

Despite this, I have a number of concerns about the plate kinematic modelling that I think need to be responded to before considering this work further, either here or for another journal:

We largely appreciate the detailed review of our manuscript done by Dr. Eagles, which derived in the identification of some crucial points regarding the production of our results that needed to be corrected. The new version of our plate kinematic model that resulted after these corrections is much more robust and can be better used to gain hints about the geodynamic processes operating along the Neogene Andean margin.

1. Statistical comparison of the new model and previous models. The model by Somoza and Ghidella is rightly chosen for the statistical comparison, because of the availability of suitable covariances for calculation of confidence regions. The uncertainties of the two methods are discussed and compared at 68% (1-sigma) level, which I note to have been done before for studies using Redback for noise reduction, but which I also note is less conservative than the 95% (2-sigma) uncertainties used in the majority of other studies in the field (including Somoza and Ghidella's). For this journal, and for reasons that I will come to, I would find the conclusions of the first test to be more convincing if the discussed differences between the two models could be proved at 2-sigma, rather than 1-sigma, level.

We agree with the reviewer's comment and therefore we compute uncertainties at the 95% confidence level. New Fig. 1 shows uncertainties in the past retro-projected position of the Nazca plate at this confidence level, and in Fig. 2 we plot convergence rate and azimuth through time using the 95% and 68% confidence levels. Both figures demonstrate that main differences between our model and Somoza and Ghidella's model also hold at the 2-sigma level, although it becomes apparent from Fig. 2 that there is a general overlap of uncertainties. We keep the 1-sigma level in this latter figure to show that, for a less conservative confidence level, differences between models do appear clearly. We also recall in the text (lines 250-255 of Changes_Noted) that main differences are observed even

at the 95% level in the time interval between 15 and 5 Ma, which is the most interesting from a geological point of view. We further concentrate our analysis in this particular time window.

2. Inferred vs. constrained rotations in the new construction. Looking at the tables for the various sets of rotations for plate pairs that have been added, I think that the high (2-3 Myr) resolution of constrained rotations (that is, a NAZ-SAM rotation that is constrained by a magnetic anomaly fitting exercise on *every* leg of the circuit used to calculate it) is interrupted over the period C5D-C5 (17.5-11.05 Ma), during which neither the ANT-AFR or AFR-SAM models of DeMets and others features any rotations. The 1-Myr stepped rotations and the changing speeds and azimuths calculated from them and displayed in Figs. 1 and 2 over this period must thus be at least partially inferred. This is important for the rest of the study, because the 17-11 Ma period envelopes and so defines one complete slow-fast-slow cycle with no synchronous named pan-Andean compressive event of its own, and in whose absence there may be no sharp slowdown to correlate with the end of the Pehuencha phase, and no sharp speed up to correlate with the start of the Quechua. In this case, the new model's results would more closely resemble those of Somoza and Ghidella, by showing a long-period slowdown in convergence rates at all studied latitudes. I am suspicious of the model in the 17-11 Ma period not only because of the absence of any pan-Andean tectonic signal, but also because of the unusually short-lived and marked ($\sim 40^\circ$) swerves in calculated NAZ-SAM azimuths. The severity of these swerves, given the stability of SAM-ANT motions and relative fixity of ANT over the same period (as noted in the manuscript's Figure 4) ought to be attributable to changes in the absolute motion of the Nazca plate, and yet appears not be reflected in the orientations of appropriately-aged segments of fracture zones formed at the flanks of the divergent NAZ-ANT plate boundary, the Chile Rise.

The ANT-AFR and AFR-SAM reconstructions of DeMets and coauthors actually do feature finite rotations in the period from 17.5 to 11.05 Ma. In fact, Table 1 in DeMets et al. (2015) for ANT-AFR and supplementary table 1 in DeMets et al. (2019) for AFR-SAM show that magnetic pickings and fracture zones identifications are available, and that these have been used to constrain the finite rotations. The sole exception is the ANT-AFR rotation since 5Dy, which we had already excluded from our analyses. In this framework, we are confident that finite rotations calculated for this time interval are as well-constrained by magnetic anomalies as those for periods before and after in the Neogene.

At the same time, we do see the point of the reviewer concerning the confidence one may have in the previous version of our reconstructions, particularly given that it featured strong, short-lived azimuth variations (17 – 15 Ma, but also at ~ 25 Ma) that are not evident in the ocean-floor fracture zones to the same extent as implied by our reconstruction. We have investigated this further and conclude that those were artefacts owing to two reasons: i) Some of the finite rotation sets utilized in this study were assigned ages by the original authors according to the geomagnetic timescale by Ogg et al. (2012), while others according to the one of Cande & Kent (1995). ii) The NZ/PA rotation at C5D is the least

constrained and most uncertain of the data set by Wilder (2003; see his Table 2). In fact, the trace of the covariance matrix for that particular rotation is significantly larger (5 to 10 times) than any other rotation in the data set. In the revised manuscript, we have therefore excluded NZ/PA C5D from our analyses, and reassigned ages to all rotations according to the timescale by Ogg et al. (2012). This eliminates the artefacts in the azimuth temporal progression but does not hamper the kinematic variations on which our inferences hinge.

Lastly, in the revised manuscript we use the most up-to-date reconstruction of the ANT-AFR motion available at the time of this revision, which has just been put forth by DeMets and coauthors (GJI, 2021). This reconstruction, which was not available at the time of our initial submission, has the advantage that it extends back to 52 Ma (as opposed to 20 Ma, as in DeMets et al., GJI 2015). It therefore eliminates the need to resort, for the period 20 to 30 Ma, to the ANT-AFR motion by Torsvik et al., 2010, which is less-resolved (10-Myr-long stages) and lacks uncertainty estimates (requiring thus an assumption made by the user). To illustrate the benefits of these steps, in the revised manuscript we added a plot of the NZ/AN and SA/AN azimuth temporal variations, in addition to their rates (new Fig. 4, lines 717-721 Changes_Noted).

To further illustrate our argumentation in this response, we add below a map of the identified fracture zones near the Chile Rise (obtained from the Global Seafloor Fabric database hosted at SOEST), overlapped with magnetic pickings by Tebbens et al., 1997 as well as age isolines from the global grid map by the EarthByte group (Seton et al., 2020). Note that in this figure, we assign ages to the anomaly pickings of Tebbens et al., 1997 according to the geomagnetic timescale of Gee & Kent 2007 (rather than Ogg 2012, which is now used throughout our reconstruction), in order to be consistent with the choice made by Seton et al., 2020 for the ocean-floor age grids (which we cannot edit). We observe that the moderate 15-20 degree change in azimuth predicted by our revised reconstruction from ~25 Ma to present day is in line with the gentle changes of direction of the few identified fracture zones that span that age interval. In our opinion this lends support to the changes (described above) that we made to our reconstruction.

3. Range of previous models. Alongside Somoza and Ghidella's, the study cites three additional previous plate kinematic works on NAZ-SAM convergence. Together, the four cited studies represent a picture of the evolution in the accuracy and resolution of models that have been available to be combined in the Nazca-Pacific-Antarctic-Africa-South America plate circuit. The new set of rotations is, in this sense, a continuation of that evolution. There is however at least one other way around the circuit that has been used, but that is not cited. This is to skip the ANT-AFR-SAM leg in favour of an ANT-SAM model derived from data in and around the Weddell Sea (<http://dx.doi.org/10.1016/j.gloplacha.2014.08.002>). Whilst the model built in this way is neither of particularly high temporal resolution nor completed with useful covariances, what it does offer is the potential to highlight consistent, and therefore robust, features in the convergence history that are at least partially independent of the detailed geometry of the circuits used.

In the revised manuscript, we now mention (lines 237-239 Changes_Noted) also the alternative reconstruction indicated by the reviewer and highlight that it yields kinematics that are in line with those of Somoza and Ghidella (2012).

Importantly, here, and referring back to my point (2), above, this alternative construction of the circuit yields convergence histories that more similar to that from Somoza and Ghidella's model than to this manuscript's one. These similarities are for a general and sustained decrease in NAZ-SAM convergence rates, and relatively unperturbed azimuths (consistent with the observable sedate changes in fracture zone azimuths on the flanks of the Chile Rise), following the onset of the Pehuenche phase around 25 Ma. With this observation, I think it becomes more important for you to

(i) be able to distinguish the new results from Somoza and Ghidella's with increased statistical rigour and, more importantly

We refer to our answer of your first comment above.

(ii) explain why the large change in NAZ-SAM azimuth at 17-11 Ma is not reflected in the shapes of Chile Rise fracture zones of a similar age,

We refer to our answer of your second comment above.

and (iii) explain why the speed-up at 15 Ma is not reflected in a whole-Andean compressive pulse, contrary to the general conclusions of your study, but seemingly at best only a local uplift near 20°S.

The new version of our model, which resulted after solving your second point, does not show an isolated pulse of convergence acceleration at 15 Ma, but a general slowdown from

16 to 11 Ma, which fits better in the tectonomagmatic evolution for the Middle Miocene along the Andes.

Minor observations:

- Abstract - I think the nature of compressive pulses is clear, but the nature of their causes and controls is not.

We agree with this comment and rephrased this text in the abstract accordingly (lines 23-24 Changes_Noted).

- Line 23 - "primary" should be 'primarily'

Fixed

- Inclusion of West-East Antarctic relative motion in the circuit. Nobody has ever done this, and yet the motion is real and may have lasted until as late as 11 Ma (<https://www.nature.com/articles/s41467-018-05270-w>).

We have taken this suggestion on board and added, both in Fig. 2 and supplementary files, an alternative reconstruction of the NZ/SA motion that includes the West-East Antarctica relative motion.

- Fig. 3 - here, I think less really could be more. I find the figure quite difficult to use due to the weight of information that is not used in the text, or material that opens more questions than it answers. For example, was there really no intrusive activity at 30°S or 20°S since 30 Ma, despite abundant evidence for volcanism?

It is very likely that volcanism was accompanied by intrusive activity at 20° and 30°S, but Neogene plutons are not exposed at these latitudes. By the contrary, at 40°S we want to highlight that magmatism, as evidenced by exhumed plutons, was active even if coeval volcanic rocks were already eroded. We think this information is important and would like to retain it in the new Fig. 3.

As well as this, the thicknesses of the blue bars in parts D, E, and F vary over a greater range than in the key. What is the significance of these changing thicknesses?

The thickness of these blue bars indicates the EW extend of compressive deformation events as informed by different authors. In this sense, thick bars at 40°S compared with much thinner ones at 20°S are mostly related to a lower level of knowledge about the localization of deformation. We think that this really helps understanding when, where and how well-constrained are the compressional events across different transects and would like to retain it in new Fig. 3. Attending the reviewer's comment, we clarify this particular point in the caption of this figure (lines 404-405 Changes_Noted).

Which of the contrasting shortening estimates shown in G and H should readers be paying most attention to?

In attention to comments of Reviewer 2, we modified these panels by: 1) replacing in G the shortening estimates from Uba et al. (2009) that were concentrated in the foreland by those

of Anderson et al. (2018) that are more updated and hold for the entire retroarc, 2) replacing in H the shortening estimates for the entire margin from Giambiagi et al. (2015) that were estimated at 33.5°S by those of Mardonez et al. (2020) that are more updated and computed right at 30°S. These modifications mean that for both transects we have now well-localized estimates at the scale of the entire orogen and for the retroarc region, which allows us doing a better comparison of the compressional deformation history across and along the strike of the margin.

Why do neither of them show any increase in shortening for the Pehuenche?

This point was already discussed in the previous version of the manuscript, particularly for the transect at 20°S where Oncken et al. (2012) indicate that this convergence acceleration is not related to an increase in the slip rate of faults but to an across-strike enlargement of the area affected by compressive deformation. Following the suggestion of Reviewer 2, in the new version we avoid referring to the Pehuenche and Diaguita phases. In addition, since we put now a much stronger focus in the time interval between 15 and 5 Ma (where our model shows the most notable differences with respect to previous plate reconstructions), this particular point is only tangentially discussed (lines 452-456 Changes_Noted).

In this last respect, I would have appreciated the introductory text being used to more clearly and rigorously define the reality and synchronicity of these phases along the Andes using up-to-date techniques and data.

This comment partially coincides with one of the main criticisms expressed by Reviewer 2, as to that the Neogene tectonic phases used in the previous version of the manuscript are out-of-use in the Andean geologic community mostly because modern geochronological data of tectono-magmatic events are hard to reconcile with strict margin-wide tectonic phases. The exception is the Quechua phase that is widely cited since there are clear evidences of a compressive tectonic event of Mid-Late Miocene age affecting the entire margin. We rewrite the introductory text considering this point and decided to not use the concept of Pehuenche and Diaguita tectonic phases in the text and figures of this new version.

- paragraph at lines 157-166: Firstly, if you do intend to keep a figure with the level of detail currently included in Figure 3, why not include these considerations of the altiplano processes there instead? Secondly, if the Juan Fernandez Ridge trajectory is not yet calculated at the resolution of the new model, why show it in Figure 3 at all?

We tried to retain in Fig. 3 only the information that we think is really important to the argumentation established in the text. In this respect, we remove for instance the track of the Juan Fernandez Ridge as suggested by the reviewer.

- line 199 . "fussing" should be 'fusing'.

Fixed

Finally, whilst I recognize that my comments are likely to mean a considerable amount of work, either for you to rebut them or to revise your models and the

manuscript, I would like to reiterate that I very much admire the rigorous approach you have taken and the high quality of the description of your work. I hope that you will be able to take and use my comments in the constructive spirit that I compiled them.

Kind regards,

Graeme Eagles

Reviewer #2 (Remarks to the Author):

The present manuscript presents an improvement in the temporal resolution of the convergence rates during the Neogene between the Nazca and South American plates. By using this improved temporal resolution, authors attempted to correlate changes in convergence rates with tectono-magmatic phases recognized along three Andean transects, which allowed them to hypothesize about the processes responsible for this correlation. However, the concept of synchronic and discrete tectonic phases in the Andes has fallen in disuse, the temporal resolution behind geological events is not comparable with the higher resolution obtained in this work and, more importantly, correlation does not mean causal relation. Although the accumulation of gravitational potential energy as a trigger for contraction is intriguing, its conversion from kinematic energy due to convergence acceleration is not supported by available data, and more work should be done in order to better understand this process. In the following paragraphs I will address each of these main concerns and some other minor issues I have detected in the present manuscript.

We sincerely thank Dr. Fennell for his in-depth and thorough review of our manuscript and the very critical points that he identified regarding the relationship between our modeled convergence velocity and the tectono-magmatic evolution of the Andean margin. We consider all the points of concern and criticisms expressed by the reviewer in his comments and the annotated PDF, modifying several aspects of the presentation and discussion of our results, as can be appreciated in the answers to these points below.

Concept of tectonic phases

When referring to the Pehuenche, Quecha and Diaguita tectonic phases, authors cite works from Mpodozis and Ramos (1989), Charrier et al. (2007), Maloney et al. (2013) and Horton (2018). However, this is a proposal made by Steinmann (1929) designed for a narrow segment, the Western Cordillera of Central Peru, that was never widely adopted for the entire Andes. It was not until the 1970s that a single group of French workers resurrected the terminology and claimed it was Andean wide. However, nothing but contradictions have arisen when geologists tried to

apply this concept throughout the Andes, making the notion of Andean building through discrete and synchronic pulses not sustainable. Of the four works cited by the authors in this manuscript, only Charrier et al. (2007) mentions one of these phases, the Quechua, but most recent works, including the compilation made by Horton (2018), do not make use of any of these terms because they are obsolete. Therefore, most readers will not be familiar with these phases and their timing, and the lack of any kind of background given by the authors makes it impossible to assess whether the results are meaningful. In the end, given the uniqueness of each sector of the Andes, this kind of simplifications and global explanations behind shortening phases always fall short.

After a detailed revision of the relevant literature mentioned by the reviewer and other available sources, we decided to consider this important criticism expressed by the reviewer and in the new version of manuscript we avoid any mention to tectonic phases other than the Quechua phase. This decision is based on: 1) our agreement with the reviewer than for most of the relevant authors there is no consensus regarding the existence of clear tectonic phases that can be valid for the entire margin at the more or less same time interval, being true that most of them even didn't talk about these phases, at least in the terms that they were originally named by Steinmann (1929) or more recent authors (Megard, 1984; Sempere et al., 1990). 2) The fact that our new results (derived from a revision of the original model after incorporation of new data and the consideration of 95% confidence intervals as suggested by Reviewer 1) show their most notable and novel features in the time interval between 15 and 5 Ma, i.e. in coincidence with the Quechua phase, which is the only one clearly recognized and explicitly mentioned by several authors (we cite now Megard, 1984; Daly et al., 1989; Sempere et al., 1990; Charrier et al. 2007 and 2013; Pfiffner et al., 2013; Gianni et al., 2017) along the entire Andean margin.

Giving this significant modification of the logical scheme used to present and discuss the results, we also modified the title (avoiding the mention to the episodic growth of the Andes), the abstract and introductory paragraphs, and figures 2 and 3 from which we remove the mention the tectonic phases.

Correlation between convergence changes and tectonic regimes

Authors propose that after an acceleration in convergence rates, the kinematic energy transmitted from the lower plate will be stored as GPE in the roots of the mountain belt, which will act as a trigger for shortening phases.

We actually do not make such an implication in our arguments. All we do is to point out, on the basis of our torque analyses, the similarity between energy stored in the mountain belt as GPE and energy removed from the budget of kinematic energy of Nazca. This, as we write in the main text, needs further investigation.

However, several parametric studies, geodynamic models and regional geological compilations of past and present deformational stages have shown that it is the

absolute motion of the overriding plate the main responsible behind the tectonic regime along the margin (Heuret and Lallemand, 2005; Lallemand et al., 2005; Sobolev and Babeyko, 2005; Schellart, 2008; van Dinther et al., 2010; Ramos, 2010; Horton, 2018, among many others).

This could be the case at a larger time-scale and under the view of low-resolution global-scale plate kinematic models. Assuming that the Antarctica plate remains relatively fixed with respect to the mantle (as shown by Torsvik et al., 2010), we demonstrate in new Fig. 4 that Neogene changes in convergence between Nazca and South America are largely due to changes in the absolute motion of the oceanic plate, with the continental plate showing a much slower and steady motion. This is also consistent with results of global-scale plate reconstructions as shown for instance by Maloney et al. (2013), something that we now mention explicitly when discussing the new Fig. 4 (lines 710-729 Changes_Noted).

Convergence speed-ups have been shown to be correlated with opening of extensional basins between 27°-46°30'S, and not with their closure (Fennell et al., 2018 and references therein).

We revised the relevant literature regarding this topic, agreeing with the reviewer about some inconsistencies in our description and argumentation. Now we explicitly indicate that convergence acceleration accompanying the birth of Nazca plate correlates with the opening of extensional basins (lines 441-443 Changes_Noted), whereas their closure via tectonic inversion seems to occur after peaking convergence rates near 20-18 Ma (lines 447-448 Changes_Noted). We cite Encinas et al. (2016 and 2021) and Fennell et al. (2018) for the ages of basin evolution. We also remark the geodynamic model of this latter author as well those of Quinteros and Sobolev (2013) and Chen et al. (2019) (both very relevant papers that we didn't included in the previous version, so thanks for the notice), to explain the connection between plate convergence, slab anchoring in the mantle transition zone and upper plate deformation (lines 448-453 Changes_Noted). We think that now it seems clear that our plate kinematic model fits into the framework presented by previous authors. In addition, we emphasize that our main focus now is the geodynamic interpretation of events occurring during the Mid-Late Miocene and not in the Late Oligocene -Early Miocene history.

Moreover, the Salar de Atacama basin at 23°S is an extensional basin developed during the late Oligocene-early Miocene convergence acceleration (Pananont et al., 2004; Jordan et al., 2007; Rubilar et al., 2018), making the claim that higher rates inhibited the opening of extensional basins in the northern transect untrue.

The Salar de Atacama basin is a very unique area of the Central Andes that shows an anomalous geologic evolution all over the Phanerozoic Era (e.g. Mpodozis and Ramos, 1989; Mpodozis et al., 2005, and references cited by you). This, added to the fact that it lies 350 km southward of our northern transect, justifies that we preferred to avoid its mention in the context of the discussion.

On the other hand, it has been shown that the deceleration of the convergence is correlated with the beginning of the Miocene contractional phase, which is produced by the difficulty of the oceanic plate when penetrating the lower mantle given the existing viscosity jump (Quinteros and Sobolev, 2013). This has led to recent proposals of slab anchoring as the trigger behind the switch from extension to compression observed along the Andes (Chen et al., 2019).

We acknowledge this point in the new version of the manuscript, as mentioned above.

Correlation between convergence changes and magmatic loci

Authors claim that convergence speed-ups correlate with eastward propagation of the magmatic arc. However, it has already been demonstrated through a great deal of works that an acceleration of convergence enhances corner flow in the asthenospheric wedge, favoring the production of magmatism at lower depths over the slab and, therefore, closer to the trench (England et al., 2004; Syracuse and Abers, 2006; England and Katz, 2010; Maunder et al., 2019; among many others). This is exactly the case during the late Oligocene – early Miocene speed-up, evidenced by the presence of widespread magmatism in the present forearc area (see Fennell et al., 2018 and references therein). Once the convergence decelerates, magmatism is produced at higher depths and, in consequence, the magmatic arc migrates eastwards. Therefore, the eastward migration of magmatism cannot be caused by convergence speed-ups, proving that correlation does not mean causal relation.

We didn't claim any causal relationship between convergence speedup and eastward propagation of the volcanic arc. We just showed that changes in convergence seem to be associated with simultaneous variations in the activity and location of magmatic arcs and fault systems, without attempting any particular geodynamic explanation at this point. In the new version of the manuscript, we tried to make clearer the mere observation of this temporal correlation.

Andean Tectono-magmatic evolution along selected transects

The compilation presented by authors is incomplete and out of date, and the most recent data supported by improved geochronological, thermochronological, sedimentological and structural studies do not correlate with the convergence accelerations obtained in the model.

In general, we do not agree with this comment because 1) we think that in the previous version we used most of the relevant available and up-to-date literature regarding the tectonic and magmatic evolution at the specific studied latitudes, and 2) the already used information plus other results included in the new version of the manuscript actually reinforce the temporal correlations previously presented.

At 20°S, Anderson et al. (2018) show that the highest shortening rates fall between 11 and 6 Ma, in contrast with Oncken et al. (2006)'s rates.

We didn't use Oncken et al. (2006) for the Altiplano shortening rates but those of Oncken et al. (2012), which are fairly comparable to those of Anderson et al. (2018). For consistency, we include now the shortening rate estimates for the retroarc region at 20°S of these latter authors in the new version of Fig. 3, replacing those of Uba et al. (2009) that were only focused in the Subandean zone. With this we show now an even better correlation between maximum shortening rates during the Late Miocene and peaking convergence rates of our model between 11 and 9 Ma.

The paleoelevation results cited by the authors are under debate, and others lines of investigation point towards a slow and steady rise of the Andes instead of growth by pulses (see Ehlers and Poulsen, 2009 and Poulsen et al., 2010), which go against the correlations and hypothesis of the authors.

Our main source of information for paleoelevation at the Altiplano transect is Garzione et al. (2014), which data at 20°S are then included in the compilation of Garzione et al. (2017). The latter is a very relevant review paper where Todd Ehlers and Chris Paulsen are co-authors. This paper supports the most updated and accepted interpretation of stable isotope in pedogenic deposits that is behind the estimation of paleoelevation. We think that this paper closed the debate mentioned by the reviewer and therefore we didn't mention it in our work.

At 30°S, Jones et al. (2016) show ongoing extension until 20 Ma, which coincides with the convergence speed-up detected in this manuscript and in previous works. This extensional stage has been recently confirmed by González et al. (2020), who interpret the end of this event occurring in the early Miocene, when convergence rates decelerated.

As explained above, we agree with the reviewer about some inconsistencies in our description of the temporal relationship between Oligocene – Early Miocene convergence speedup and opening of extensional basins at 40°S and 30°S. As we now concentrate our analysis in the Mid-Late Miocene phase, we briefly describe this relationship (lines 441-443 Changes_Noted).

Shortening rates at 30°S recently published by Mardonez et al. (2020) differ from the ones estimated by Allmendinger and Judge (2014), showing that the highest rates occurred between 8 and 5 Ma, uncorrelated with any of the phases proposed in the manuscript.

We didn't included the shortening rates estimated by Mardonez et al. (2020) in the previous version because they were not reported by the authors. Nevertheless, following the reviewer's comment, we calculated shortening rates dividing the estimated total shortening by the duration of time intervals reported in Table 1 of Mardonez et al. (2020), and include them in the new version of Fig. 3 (replacing those of Giambiagi et al., 2017 that were similar to Mardonez et al., 2020 but estimated at 33.5°S). We note now that peak shortening rates in the retroarc at 30°S do occur between 10 and 8 Ma, in an almost perfect coincidence with maximum shortening rates at the Precordillera reported by Allmendinger and Judge (2014) and peak convergence rates of our model between 11 and 9 Ma.

It is surprising that the authors do not compile any information at 40°S, given the wealth of published articles in this area (see Echaurren et al., 2019 and references therein).

We do actually compile information at 40°S. In fact, Fig. 3F is based on a large compilation of recently published papers around 40°S, as reported in the Supplementary Material. Particularly, we used in the previous version the papers of Orts et al. (2012, 2015), Garcia-Morabito et al. (2011, 2012), Echaurren et al. (2016), Horton (2018), Encinas et al. (2018), Bechis et al. (2014), Cembrano et al. (2002), Navarrete et al. (2020). For this revised version of the manuscript, we incorporated other papers suggested by the reviewer, but only those focused in study areas that are at most one degree away from 40°S, such as Ramos et al. (2015) and Fernandez Paz et al. (2019). We found that information contributed by these papers fits in the scheme already drawn for Fig. 3F. We chose not to use data collected more than one degree southward than 40°S (like Echaurren et al., 2019 or Folguera et al., 2018), because of the complex and strongly varying tectonic structure and evolution of the Patagonian broken foreland. Unfortunately, we were unable to find any published estimate of shortening rates at 40°S.

For example, Ramos et al. (2015) and Folguera et al. (2018) propose several shortening events during the Neogene, but none of them coincide with the acceleration in convergence rates detected in this work.

This is incorrect: the tectonic evolution proposed by Ramos et al. (2015) is indeed very consistent with the picture already gave in the original version of Fig. 3F. For the new version we slightly corrected the spatio-temporal location of deformation events as informed by these authors at 41°S. The paper of Folguera et al. (2018) is focused in an area between 45°-47°S, too far away from our transect to be considered.

Moreover, extensional deformation at these latitudes overlaps with high convergence rates (see Fernandez Paz et al., 2019), in contrast with the hypothesis developed in this manuscript.

We include this paper when discussing the relationship between extensional basin development and convergence acceleration during Oligocene – Early Miocene times, as mentioned above (lines 443-445 Changes_Noted).

In conclusion, if authors made a more exhaustive and complete compilation of the Andean tectonic phases at the selected latitudes, they would find that there are no correlations, that the timing between both datasets is not comparable, or that a lag time might exist between speed ups and shortening. The only existing correlation supported by actual data is a switch between a brief and widespread extensional event towards a shortening phase that occurred in the early Miocene, although this is not something synchronic along the margin. The existence of “sub-phases” within the Miocene contractional stage is not supported by any of the most recent datasets supported by modern geo- and thermochronological analyses.

Our revised results in terms of changes in convergence rate and the detailed revision of geologic information that was necessary to perform in order to answer the comments of Dr. Fennell, allowed us to refine our interpretations and support our conclusions in a clearer and stronger way.

Minor issues

Although I agree that the 50% difference between present convergence rates and GPS measurements detected by authors could be attributed to the megathrust cycle, I encourage authors to elaborate on this subject and run tests of their model in other settings devoid of this kind of cycles in order to test its validity.

Thanks for the suggestion, we will consider it when developing a detailed analysis of this topic in the near future.

Given that it is the absolute motion of the overriding plate the first order parameter controlling the tectonic regime, taking the South American plate as an almost absolute reference frame seems odd. Even small differences (extension switched to compression after the opening of the Atlantic Ocean at low rates) might affect the overall tectonic regime, something worth investigating.

We didn't say that South America (SA) is an almost absolute reference frame, instead we demonstrate that the relative motion between SA and Antarctica (AN) is much smaller than the motion between AN and Nazca (NZ). Given that AN behaves almost as a fixed plate with respect to the mantle during the Neogene (i.e. Torsvik et al., 2010), we can argue that changes in NZ-SA convergence are mostly due to the motion of the NZ plate, with the absolute motion of SA having a much smaller effect. We clarify this point in the new version (lines 708-710 Changes_Noted) in order to avoid possible confusions.

The idea of delamination occurring in the northern transect affecting the whole Andean margin is speculative and unsupported, and authors should elaborate more before making this proposal. What happened to the north of 20°S? Would

this process be synchronic or evolve gradually? Are there any models or works that support this idea?

This is certainly a provocative idea that is supported by our results into the conceptual framework provided by our torque balance calculations; we would like to retain this idea in the manuscript because it can stimulate an interesting debate about the role of geodynamic processes occurring essentially to the upper plate in regulating the motion of converging plates. Although integrating geologic information to the north of 20°S could be interesting and useful, we think it should be reserved for another work where this very topic could be deeply explored.

I will be glad to clarify any of my comments and share any of the papers referenced in the revision and attached PDF.

Sincerely,

Dr. Lucas Fennell

References to papers mentioned in the revision not included in main manuscript:

- Heuret and Lallemand, 2005. Plate motions, slab dynamics and back-arc deformation.
- Lallemand et al., 2005. On the relationships between slab dip, back-arc stress, upper plate absolute motion, and crustal nature in subduction zones.
- Sobolev and Babeyko, 2005. What drives orogeny in the Andes?
- Schellart, 2008. Subduction zone trench migration: slab driven or overriding-plate-driven?
- van Dinther et al., 2010. Role of the overriding plate in the subduction process: Insights from numerical models.
- Ramos, 2010. The tectonic regime along the Andes: Present-day and Mesozoic regimes.
- Fennell et al., 2018. The role of the slab pull force in the late Oligocene to early Miocene extension in the Southern Central Andes (27-46 S): Insights from numerical modeling.
- Pananont et al., 2004. Cenozoic evolution of the northwestern Salar de Atacama Basin, northern Chile.
- Jordan et al., 2007. Cenozoic subsurface stratigraphy and structure of the Salar de Atacama Basin, northern Chile.
- Rubilar et al., 2018. Structure of the Cordillera de la Sal: A key tectonic element for the Oligocene-Neogene evolution of the Salar de Atacama basin, Central Andes, northern Chile.
- Quinteros and Sobolev, 2013. Why has the Nazca plate slowed since the Neogene?
- Chen et al., 2019. Southward propagation of Nazca subduction along the Andes.
- England et al., 2004. Systematic variation in the depths of slabs beneath arc volcanoes.
- Syracuse and Abers, 2006. Global compilation of variations in slab depth beneath

arc volcanoes and implications.

- England and Katz, 2010; Melting above the anhydrous solidus controls the location of volcanic arcs.
- Maunder et al., 2019. Modeling slab temperature: A reevaluation of the thermal parameter.
- Ehlers and Poulsen, 2009. Influence of Andean uplift on climate and paleoaltimetry estimates.
- Poulsen et al., 2010. Onset of convective rainfall during gradual late Miocene rise of the central Andes.
- Jones et al., 2016. The role of changing geodynamics in the progressive contamination of Late Cretaceous to Late Miocene arc magmas in the southern Central Andes.
- González et al., 2020. The Iglesia basin in the southern Central Andes: A record of backarc extension before wedge-top deposition in a foreland basin.
- Mardonez et al., 2020. The Jáchal river cross-section revisited (Andes of Argentina, 30° S): Constraints from the chronology and geometry of neogene synorogenic deposits.
- Echaurren et al., 2019. Tectonic controls on the building of the North Patagonian fold-thrust belt (~ 43° S).
- Ramos et al., 2015. The North Patagonian orogenic front and related foreland evolution during the Miocene, analyzed from synorogenic sedimentation and U/Pb dating (~ 42° S).
- Folguera et al., 2018. Constraints on the Neogene growth of the central Patagonian Andes at the latitude of the Chile triple junction (45–47 S) using U/Pb geochronology in synorogenic strata.
- Fernandez Paz et al., 2019. Constraints on trenchward arc migration and backarc magmatism in the north patagonian andes in the context of nazca plate rollback.

All of these papers were revised and most of them included in the new version, mostly those contributing with data and ideas that help to understand the tectono-magmatic evolution at or near the studied transects.

Reviewer #3 (Remarks to the Author):

This is a very good paper, addressing a fundamental aspect of crustal evolution, namely the growth of Andean-style mountain belts and possible links to global plate motion, especially convergence velocity. The more detailed kinematic plate motion model presented by the authors is an important step in hypothesis testing. Some minor wording changes would help to clarify the arguments (see attached document) and I have a few more substantive comments on figures that could also help to clarify their arguments (below).

We thank Reviewer 3 for the helpful comments and suggestions expressed in his corrected PDF document (from which we incorporated almost all of them in the new version) and those answered below.

The Introduction nicely sets the stage for the paper, reviewing past work including papers that have explicitly linked slowing convergence to shortening via mountain growth and augmented frictional resistance on the megathrust. In this regard the 1999 paper by Norabuena (GRL 26) was I believe one of the first and perhaps should be referenced.

Thanks for the notice; we included Norambuena et al. (1999) when describing the hypothesis that connect convergence slowdown and mountain growth via frictional resistance at the megathrust (line 77 Changes_Noted).

Figure 1 shows retro-projected points, but despite a very long caption, never says what the points represent. Are these points evenly spaced in time (e.g, 1 million years) so that speed variations can be inferred from point spacing? If not, what do the points represent?

We explicitly mention in the caption of new Fig. 1 (lines 176-177 Changes_Notes) that these points are the retro-projected positions of points currently entering the trench as predicted by our model and Somoza and Ghidella (2012).

There are also some labels on South America, but they are difficult to read against the color background. The geologic details shown are not necessarily important to the arguments presented. I suggest the authors revise this figure to focus on things that are important to their argument.

We follow the recommendation of the reviewer and replace the geologic map (that was actually unnecessary) with a topographic/bathymetric map of the Andean margin, which allow us to better describe difference in morphology that are related with total shortening. We maintain the labels on morphostructural units because they are related to Fig. 3.

Figure 3 is similarly very "busy" and could be improved. The convergence rate plots (left hand panels) show a high rate and a low rate line, which presumably represent trench-parallel and trench-perpendicular, but on my print out both come out as dark lines. I suggest making the trench-parallel one either much lighter, or dashed. Alternately consider omitting it - is it necessary?

We modified the intensity of grey tones for trench-perpendicular and trench-parallel component of convergence in order to better distinguish between them. We prefer to maintain the latter because it shows significant temporal variations that are associated to the tectonic evolution.

Related to Figure 3, it seems to me a key point of the paper is a correlation between convergence rate between the plate and shortening rate in the upper plate. Perhaps a scatter plot comparing these two quantities would be useful to explicitly show the connection.

This could have been possible if published shortening rates were similarly obtained at different latitudes and with similarly high temporal resolution than our convergence estimates. Due to one comment of Reviewer 2, in the new version of Fig. 3 we included for instance the estimate of orogen-wide shortening rates of Anderson et al. (2018) that in the detail are somehow different than Oncken et al. (2012), although the main conclusions stand. So, we prefer to keep the comparison between convergence and shortening rates just in a qualitative sense, without forcing an unjustified quantitative approach.

REVIEWER COMMENTS

Reviewer #1 (Remarks to the Author):

Dear authors,

thankyou for taking the time and considerable effort to respond to my comments on the first version of your manuscript.

Those comments focussed on two main areas of the manuscript: the plate kinematic modelling, and the orogenic record. As you'll remember, I was intrigued to see that it might be possible to link the two using an attractively-simple torque model.

This time around, I will organize my review in a similar way to the first one.

Plate Kinematic Modelling:

I must apologize that one of my first-round comments about the plate kinematic modelling (concerning the temporal resolution of the input models) was not well phrased, and consequently too easy to misunderstand. Clearly, I agree with you that the very detailed work of DeMets and others included numerous rotations in the 17.5-11.05 Ma period. My comment really ought to have made obvious that it is the models by Croon et al, and by Wilder, that are differently- and/or more-coarsely sampled, over that period. As the thorough re-evaluation of your model in the light of my other comments showed, the addition of non-synchronous sets of rotations, which involves interpolation to the times of interest, can lead to undesirable results. One crass reason for non-synchronicity is if model timescales are not harmonized prior to interpolation (as you found for this revision, the artefacts can be large and appear interpretable). Even after harmonization, however, artefacts can persist, for example, when a long interpolated period happens to bridge over, and thus is unable to precisely resolve the effect of, a short-lived change in plate divergence.

In its new version, your kinematic model now (i) more closely resembles its lower-resolution predecessors and (ii) does not imply implausibly-drastic plate motion changes across the Chile Rise. A significant consequence of this is that one of the first version's multiple acceleration-deceleration cycles is shown to have been entirely spurious (you state in your response letter that this was owed to a missed timescale harmonization). In the revised model, two such cycles remain, separated by a short (~14-11 Ma) dip in convergence rates. Without this dip, your model would show very strong long-period similarity to that of Somoza and Ghidella and the other older low-resolution models of Nazca-South America convergence. In light of the general expectation that tectonic plates have great momentum, short-period changes like the dip are remarkable signals; they can be understood in terms of remarkable changes in plate driving forces, or alternatively as irritating artefacts. Having seen the vulnerability of the circuit construction to interpolation-related artefacts demonstrated as part of the revision of your first version of the model, we might now be justified in similarly suspecting the 14-11 Ma dip to be a possible artefact. This suspicion is not baseless because (i) the 14-11 Ma period is bridged in Croon et al's model by 4 rotations, in Wilder's by 3, and in the DeMets models by 4, with the Croon et al and Wilder rotations being differently-spaced in time than both each other and the DeMets models and (ii) the dip accompanies a sharp (~7-15°) rotation in convergence azimuth, reminiscent (albeit by no means as spectacular) of the changes accompanying the now-discounted Pehuenche acceleration-deceleration cycle from the first version of your model.

Based on this, I think it is now important for you to convince your readers that the dip at 14-11 Ma is not an artefact, or (if it is one) to alter your research question, model, and manuscript accordingly. The easier way to approach this might be to interrogate your model circuit in such a way that it can be used to predict the shapes of the Chile Rise fracture zones over the 14-11 Ma period. As we agreed for the previous, larger, artefact in the first model version, it can be expected that the NAZ-SAM azimuth change at 14-11 Ma, if it is not artefactual, should be reflected in those shapes. An alternative, involving much more effort, would be to generate a new high-resolution model, or models, to replace those of Wilder and Croon et al, in your circuit constructions.

Orogenic record:

Some of my other comments concerned the observational reality of the multiple Andean convergent cycles that version 1 of the manuscript dealt with. Thanks to the review by Dr Fennell, and your responses to it, it seems now clear to me that the occurrence of multiple cycles is also artefactual. Instead of the situation described in the first version of your manuscript, it seems that only one 'event', the Quechua, may be considered observationally significant at the resolution currently available to us. This leaves me wondering to what extent the Quechua is representative of global (or even Andean) convergence processes, and hence to what extent the attempt to model it in terms of the exchange of convergent and gravitational forces (as you set out to do) was really worth trying. Early on in the revised manuscript, you cite plenty of studies about the Quechua, but just why and how significant or representative it is, or even how precisely it can be dated compared to the resolution of your plate kinematic model, is not clearly stated. A figure would be the ideal vehicle to show what it is about the Quechua that makes it suitable as the subject of your study. The closest you get to such a figure is Fig. 3, but this remains difficult to divine in the face of the sheer mass of information presented. That figure is an admirable work of compilation, and no doubt will be useful elsewhere, but even after the removal of the 'Juan Fernandez Ridge' it remains an ineffective messenger in the context of your manuscript. For now, I note again that I am no expert in Andean geology, but it may be significant that at least Dr Fennell's publications, which are presumably representative of the state of the art in Andean orogenic studies, seem not to use the 'Quechua' term at all.

To summarize, at the moment the manuscript's residual attraction to me lies in the intuitive appeal of the results of the first-order torque balance modelling (only two of very many forces are considered). By now, however, that appeal is outweighed by justifiable doubts about the manuscript's starting assumptions and, to an extent, its results. Specifically, first, is the Quechua a real example of a real, significant, widespread, phenomenon of "orogenic cyclicity" and, second, does the plate kinematic model really reliably resolve the continuity of Quechua-period motion at the 1-Myr level of resolution (especially in the 14-11 Ma period)? In the first instance, I am unable to offer much constructive feedback, other than to pay attention to the presentation of the message in Figure 3. In the second, I think there are ways ahead to test the reality, or otherwise, of the 14-11 Ma dip that makes your plate kinematic model, in its current form, significantly different than its predecessors in the literature.

Best regards,

Graeme Eagles

Reviewer #2 (Remarks to the Author):

After reading the present manuscript, which constitutes an in-depth revision of the previous version, I conclude that the changes made by the authors have improved the overall flow and quality of the manuscript. The decision of going for a unique construction event during the Neogene instead of a series of episodes falls in line with current datasets and interpretations, and matches well with the improved high-resolution convergence rates. I believe that this is the result of a positive exchange between reviewers and authors, which is reflected by the thoroughness of the rebuttal letter, where I gladly observed that my and the other reviewers concerns, suggestions and criticisms were seriously considered and addressed.

In this line, one of my main concerns in the previous version was that the convergence acceleration coincided with the opening of a series of extensional basins, an observation the authors now agree with. However, in line 180, they suggest that no extensional basins opened at 20°S because convergence was higher there than southward, which falls in contradiction with the change in their interpretation. Moreover, in their figure 3, they clearly show that the rates were higher at 30°S and 40°S during the earliest Miocene,

which makes their claim unsustainable. On the other hand, I am happy to see that the authors have acknowledged slab anchoring at 660km as a potential mechanism behind the switch from extension to compression throughout the Andes. I also appreciate that the authors consider the motion of the upper-plate as dominant at a larger time-scale, whereas pulses of deformation can be correlated with changes in convergence related to the motion of the oceanic plate within a continuous contractional regime, such as the one the Andes have been experiencing since Cretaceous times.

Regarding the Andean tectono-magmatic evolution, I must admit that some of my criticisms to the previous version were exaggerated, although my intention was to show that the interpretation of the entire Andes as being built through three synchronous and perfectly timed deformational phases was incorrect. The modification of this interpretation and the clarification that no causal relation is claimed between magmatism and convergence rates has reassured me. Although the magmatic loci and the variations in convergence rates show some degree of correlation, I consider that the geodynamic explanation behind this fact deserves further exploration and debate. Updated shortening rates at 20°S and 30°S adjust well with the new results and, despite the unfortunate non-existence of published rates at 40°S, I believe that the information compiled for Patagonia points towards a similar direction. Regarding the paleoelevation of the Puna plateau, I agree with the authors that the available datasets suggest an episodic uplift, although this debate is far from being closed.

In summary, I consider that the present manuscript constitutes a fine contribution to Nature Communications, and that the high-resolution convergence rates will be useful to address both present and future tectonic and geodynamic debates. I must also admit that the mechanism proposed for convergence variations and tectono-magmatic changes in the upper plate is intriguing, now even more than in the previous version. Therefore, I look forward to seeing the next steps of this project and the further exploration of the delamination hypothesis.

Sincerely,
Dr. Lucas Fennell

Reviewer #3 (Remarks to the Author):

The revised manuscript answers almost all of the reviewer concerns and in my opinion is ready for publication. I found a few minor wording issues in the abstract and text that the authors may wish to change, but all are minor and at the authors' discretion (referenced by Line #):

23 'emerging from' could be "as observed in new"

28 'to revise' could be "revision of"

30 'to' could be "with"

35 'result of hundred' could be "result of one hundred"

63 'hampered the possibility to establish' could be "hampered establishment of"

75 'allow' should be "allows"

92 'benefits' should be "benefit"

(92 - a sentence describing the utility of Redback might be useful here)

98 "of points on the Nazca plate"

116 'very' should be "more"

121 'does occur' could be "occurs"

210 'to' should be "with"

213 'for' should be "to"

214 While "fusing is formally correct here, applying it to the lower and middle crust will imply to some readers that you are proposing to join together the lower and middle crust. Using the verb "melting" would avoid this ambiguity.

217 'drops should be "drop"; also I would omit "down"

222 'with' could be "to"

REBUTTAL LETTER FOR THE SECOND REVIEW OF THE MANUSCRIPT:

Growth of the Neogene Andes linked to changes in plate convergence using high-resolution kinematic models

We greatly appreciate the revision of the second version of our manuscript done by each reviewer. Particularly significant were comments expressed by Reviewer 1 regarding the reality of convergence changes implied by our results in the period between 15 and 10 Ma. We have carefully considered his comments and suggestions, performing new tests that allow us to demonstrate the reliability of this very feature of our model. Comments of Reviewers 2 and 3 were very positive to this second version and only minor points raised by them are now included in this third version. The document Changes_Noted include all the modifications done to the manuscript.

Below we tried to isolate specific comments/suggestions of each reviewer in order to give an answer (in blue) to all of them. For most of these answers, we identified the line numbers in the document Changes_Noted where the implied modifications were included.

We think that all these changes really help improving the quality of our work and its potential impact. We hope that the Editor and Reviewers will agree with us in this respect.

REVIEWER COMMENTS

Reviewer #1 (Remarks to the Author):

Dear authors,

thank you for taking the time and considerable effort to respond to my comments on the first version of your manuscript.

Those comments focused on two main areas of the manuscript: the plate kinematic modelling, and the orogenic record. As you'll remember, I was intrigued to see that it might be possible to link the two using an attractively-simple torque model.

This time around, I will organize my review in a similar way to the first one.

Plate Kinematic Modelling:

I must apologize that one of my first-round comments about the plate kinematic modelling (concerning the temporal resolution of the input models) was not well phrased, and consequently too easy to misunderstand. Clearly, I agree with you that the very detailed work of DeMets and others included numerous rotations in the 17.5-11.05 Ma period. My comment really ought to have made obvious that it is the models by Croon et al, and by Wilder, that are differently- and/or more-coarsely sampled, over that period. As the thorough re-evaluation of your model in the light of my other comments showed, the addition of non-synchronous sets of rotations, which involves interpolation to the times of interest, can lead to undesirable results. One crass reason for non-synchronicity is if model timescales are not harmonized prior to interpolation (as you found for this

revision, the artefacts can be large and appear interpretable). Even after harmonization, however, artefacts can persist, for example, when a long interpolated period happens to bridge over, and thus is unable to precisely resolve the effect of, a short-lived change in plate divergence.

In its new version, your kinematic model now (i) more closely resembles its lower-resolution predecessors and (ii) does not imply implausibly-drastic plate motion changes across the Chile Rise. A significant consequence of this is that one of the first version's multiple acceleration-deceleration cycles is shown to have been entirely spurious (you state in your response letter that this was owed to a missed timescale harmonization). In the revised model, two such cycles remain, separated by a short (~14-11 Ma) dip in convergence rates. Without this dip, your model would show very strong long-period similarity to that of Somoza and Ghidella and the other older low-resolution models of Nazca-South America convergence. In light of the general expectation that tectonic plates have great momentum, short-period changes like the dip are remarkable signals; they can be understood in terms of remarkable changes in plate driving forces, or alternatively as irritating artefacts. Having seen the vulnerability of the circuit construction to interpolation-related artefacts demonstrated as part of the revision of your first version of the model, we might now be justified in similarly suspecting the 14-11 Ma dip to be a possible artefact. This suspicion is not baseless because (i) the 14-11 Ma period is bridged in Croon et al's model by 4 rotations, in Wilder's by 3, and in the DeMets models by 4, with the Croon et al and Wilder rotations being differently-spaced in time than both each other and the DeMets models and (ii) the dip accompanies a sharp (~7-15°) rotation in convergence azimuth, reminiscent (albeit by no means as spectacular) of the changes accompanying the now-discounted Pehuenche acceleration-deceleration cycle from the first version of your model.

Based on this, I think it is now important for you to convince your readers that the dip at 14-11 Ma is not an artefact, or (if it is one) to alter your research question, model, and manuscript accordingly. The easier way to approach this might be to interrogate your model circuit in such a way that it can be used to predict the shapes of the Chile Rise fracture zones over the 14-11 Ma period. As we agreed for the previous, larger, artefact in the first model version, it can be expected that the NAZ-SAM azimuth change at 14-11 Ma, if it is not artefactual, should be reflected in those shapes. An alternative, involving much more effort, would be to generate a new high-resolution model, or models, to replace those of Wilder and Croon et al, in your circuit constructions.

We are grateful to the reviewer for the time to evaluate our submission and for his comments. We have carefully considered them and tested our reconstruction in light of them. From our analyses, which we illustrate in detail below, we conclude that the kinematic pattern between 14 and 11 Ma visible in our reconstruction is not an artefact arising from interpolation, but a true feature of the data. The kinematic pattern can be verified against independent observations of the fracture zones orientations (as suggested by the reviewer) as well as of the Chile Rise spreading history, which has been inferred directly from observed magnetic lines. Furthermore, in the final part of our response we illustrate the reasons why our reconstruction differs from that of Somoza & Ghidella, 2012 (SG12) around the time of the Quechua deformation phase.

We agree with the reviewer on the appropriateness of convincing the readership of the robustness of the novel feature of our reconstruction (i.e, the kinematic pattern between 14 and 11 Ma). Having harmonized timescales among the finite-rotation sets eliminates the issues of large fluctuations in the temporal pattern of convergence. We believe the reviewer will agree with us that interpolation is a necessary and valuable tool to combine differently-resolved data sets, without losing any of the information contained in the more-resolved sets among them. Interpolation in itself is not a procedure that would generate large fluctuations in convergence temporal patterns (when performed on properly-harmonized sets of rotations), because it can only yield a result in between two data-constrained values that lie to the left- and right-hand sides of the interpolated point (on a time line, in this case). At the same time, we agree with the reviewer that it is worth testing whether some artefacts still persist upon interpolation, albeit smaller than those associated with non-harmonized timescales. New figure 4 in the main text illustrates that the kinematic pattern between 14 and 11 Ma shown in our reconstruction ultimately comes from the Nazca-Pacific-Antarctica part of the plate circuit. Therefore, in order

to convince the readers that such a pattern is not an artefact, we have performed three additional interpolations of this part of the circuit using three sets of ages: i) that of the data set of Wilder 2003, ii) that of the data set of Croon et al., 2008, and iii) that of an independent reconstruction of the Nazca/Antarctica spreading history by Tebbens et al. (1997), which is based on 17 magnetic lines observed across the Chile Rise. Results are shown in Supplementary Figure S3. The fact that the 14-11 Ma deceleration/acceleration pattern of the Nazca/Antarctica motion remains visible regardless of the specific set of ages utilized for the interpolation indicates that such a pattern is a true feature of the data, rather than an artefact arising from the specific choice of ages.

There is a body of work, carried out by S. Tebbens, S. Cande and co-authors, that constitutes an independent constraint to our reconstruction of the Nazca/Antarctica motion. They relied on a magnetic survey of the Chile Rise that collected a total of 19 magnetic lines crossing the spreading ridge. From these, they inferred the spreading history of the Chile Rise back to ~24 Ma at a resolution of 2-3 Myr, which is presented in S. Tebbens' PhD thesis as well as in Tebbens et al., JGR 1997 (TA97 from here on). In a companion paper (Tebbens & Cande, JGR 1997 – TC97 from here on) published in the same volume of TA97, authors use the magnetic lines to obtain Nazca-Antarctica finite rotations only back to ~12.2 Ma. Authors also extrapolated the 12.2 Ma finite rotation further back to ~16 Ma, in order to temporally connect their reconstruction to rotations along the Farallon-Pacific-Antarctica from ~20 Ma to ~33 Ma. It is important to make this clarification about the literature and the specific inferences drawn by these authors (spreading history back to ~22 Ma and finite rotations only back to ~12 Ma) because here we use TA97 as an independent constraint to test the novel feature of our reconstruction, whereas SG12 used a selection of rotations from TC97 (further details below). In Figure 4 of the manuscript and Supplementary Figure S3 we compare our reconstruction of the Nazca/Antarctica motion (interpolated using several sets of ages) to the Chile Rise spreading history by TA97. We consider the latter to be a prime constraint because the spreading history is inferred directly from a wealth of magnetic lines over a latitudinal span of ~10 degrees, at relatively high temporal resolution (please see Table 2 and Discussion in TA97). The fact that our reconstruction agrees with that of TA97, particularly in the interval 14-11 Ma and when we interpolate it at their resolution (see blue and magenta in Supplementary Figure S3a) lends support to the notion that our kinematic pattern is a true feature of the data. Furthermore, our reconstruction is in line with a change in azimuth along the Chiloe and Guafo fracture zones (FZs) documented by TA97. In addition to the constraint above, and following the reviewer suggestion, we test the change in azimuth between ~14 and ~12 Ma predicted by our reconstruction of the Nazca/Antarctica motion against the shape of FZs mapped around the Chile Rise from vertical gravity gradients (VGG, Matthews et al., 2011). This is now described in the Methods section and illustrated in Supplementary Figure S4. We find that a general trend of change in azimuth is visible in the mapped FZs, but remain aware of the limits dictated by the resolution and uncertainty of the VGG-mapped FZs (see Methods section). As we believe that these tests are an important contribution to demonstrate the feasibility of our results, mostly for the time interval when we observe the largest discrepancies with previous authors, we now include this in the main text (lines 360-387 of Changes_Noted) with references to Supplementary Figures S3 and S4.

We also added a paragraph to the Methods section called *Focusing on the Nazca/Antarctica motion* where we describe the technical details involved in the tests mentioned above (lines 509-542 Changes_Noted). Finally, we included a paragraph in the main text (lines 164-179 Changes_Noted) to describe the reasons why the 14-11 Ma deceleration/acceleration pattern of Nazca visible in our reconstruction is not present in SG12. This owes to the specific selection of Nazca-Antarctica finite rotations that SG12 made from the set of TC97. Here we feel it is important to recall that the magnetic data across the Chile Rise used by Tebbens, Cande and co-authors have been used to infer i) the spreading-rate history back to ~24 Ma (published in T97), but ii) finite rotations only back to ~12 Ma (published in TC97). In fact, quoting from the introduction to S. Tebbens' PhD thesis, which preceded the two publications, 'In Chapter 3, the NRL-LDEO 1990 aeromagnetism survey of the Chile ridge are examined to determine the spreading rate history of the Chile ridge for the past 24 million years and the Nazca-Antarctic poles of rotation for the past 12 million years'. TC97 performed two interpolations and one extrapolation (from ~12 to ~16 Ma) of the data-constrained finite rotations (please see Table 4 in TC97, and in particular the Reference column and bottom notes). SG12 did not use all these finite rotations, but a selection of only three rotations that included one data-constrained at 4.9 Ma, one interpolated at 10.8

Ma, and one extrapolated at 16 Ma (please see the second table on page 1 of the Supplementary Data of SG12). This choice effectively means that in the reconstruction of SG12 steadiness of the Nazca/Antarctica motion between 16 and 11 Ma is enforced, despite the data constraints were only back to ~12 Ma. Thus, by construction SG12 cannot resolve any changes in the Nazca motion during the period of time from 16 to 11 Ma. We believe this is an important point that will help readers to understand the real source of difference between our model and SG12.

Orogenic record:

Some of my other comments concerned the observational reality of the multiple Andean convergent cycles that version 1 of the manuscript dealt with. Thanks to the review by Dr Fennell, and your responses to it, it seems now clear to me that the occurrence of multiple cycles is also artefactual. Instead of the situation described in the first version of your manuscript, it seems that only one 'event', the Quechua, may be considered observationally significant at the resolution currently available to us. This leaves me wondering to what extent the Quechua is representative of global (or even Andean) convergence processes, and hence to what extent the attempt to model it in terms of the exchange of convergent and gravitational forces (as you set out to do) was really worth trying. Early on in the revised manuscript, you cite plenty of studies about the Quechua, but just why and how significant or representative it is, or even how precisely it can be dated compared to the resolution of your plate kinematic model, is not clearly stated. A figure would be the ideal vehicle to show what it is about the Quechua that makes it suitable as the subject of your study. The closest you get to such a figure is Fig. 3, but this remains difficult to divine in the face of the sheer mass of information presented. That figure is an admirable work of compilation, and no doubt will be useful elsewhere, but even after the removal of the 'Juan Fernandez Ridge' it remains an ineffective messenger in the context of your manuscript. For now, I note again that I am no expert in Andean geology, but it may be significant that at least Dr Fennell's publications, which are presumably representative of the state of the art in Andean orogenic studies, seem not to use the 'Quechua' term at all.

In order to answer this point, we refer to the second review of Dr. Fennell and most specifically to his comments regarding our choice to concentrate the analysis in just one Neogene tectonic phase (Quechua): *The decision of going for a unique construction event during the Neogene instead of a series of episodes falls in line with current datasets and interpretations, and matches well with the improved high-resolution convergence rates.*

He also wrote:

Regarding the Andean tectono-magmatic evolution, I must admit that some of my criticisms to the previous version were exaggerated, although my intention was to show that the interpretation of the entire Andes as being built through three synchronous and perfectly timed deformational phases was incorrect. The modification of this interpretation and the clarification that no causal relation is claimed between magmatism and convergence rates has reassured me.

In contrast to the (Late Oligocene – Middle Miocene) Pehuenche and (Late Miocene – Pliocene) Diaguita phases that were defined based on old data and were no longer used in the Andean literature since the mid '90s, the Mid Miocene Quechua phase is indeed clearly recognized and explicitly mentioned by several modern studies, most of them being now included in the introduction and discussion of our manuscript. Furthermore, and as indicated also by the reviewer, we have done a large effort compiling the relevant information integrated in Fig. 3 where we think that now it is much clear to observe the temporal correlations between convergence acceleration between 12 and 9 Ma and tectono-magmatic changes that jointly define the Quechua phase.

To summarize, at the moment the manuscript's residual attraction to me lies in the intuitive appeal of the results of the first-order torque balance modelling (only two of very many forces are considered). By now, however, that appeal is outweighed by justifiable doubts about the manuscript's starting assumptions and, to an extent, its results. Specifically, first, is the Quechua a real example of a real, significant, widespread, phenomenon of

"orogenic cyclicality" and, second, does the plate kinematic model really reliably resolve the continuity of Quechua-period motion at the 1-Myr level of resolution (especially in the 14-11 Ma period)?

We hope that tests and arguments put forth in this revision can satisfactorily convince the reviewer that the answer to both questions is "Yes".

In the first instance, I am unable to offer much constructive feedback, other than to pay attention to the presentation of the message in Figure 3. In the second, I think there are ways ahead to test the reality, or otherwise, of the 14-11 Ma dip that makes your plate kinematic model, in its current form, significantly different than its predecessors in the literature.

Best regards,

Graeme Eagles

We would like to thank again Dr. Eagles for the very rigorous and constructive review of our manuscript. The additional work necessary to answer his comments and criticism indeed helped identifying spurious artefacts of the plate kinematic model in the first review and consolidating the feasibility and credibility of our results in this second round.

Reviewer #2 (Remarks to the Author):

After reading the present manuscript, which constitutes an in-depth revision of the previous version, I conclude that the changes made by the authors have improved the overall flow and quality of the manuscript. The decision of going for a unique construction event during the Neogene instead of a series of episodes falls in line with current datasets and interpretations, and matches well with the improved high-resolution convergence rates. I believe that this is the result of a positive exchange between reviewers and authors, which is reflected by the thoroughness of the rebuttal letter, where I gladly observed that my and the other reviewers concerns, suggestions and criticisms were seriously considered and addressed.

We gratefully acknowledge the original comments and constructive criticism of Dr. Fennell that forced us to re-evaluate the conceptual framework used to interpret our plate kinematic results in relation to the Andean tectono-magmatic evolution. We also largely agree with his statement that the last version of the manuscript, after incorporating his suggestions and those of the other reviewers, is the result of a very positive exchange between reviewers and authors that greatly improved the clarity and quality of our work.

In this line, one of my main concerns in the previous version was that the convergence acceleration coincided with the opening of a series of extensional basins, an observation the authors now agree with. However, in line 180, they suggest that no extensional basins opened at 20°S because convergence was higher there than southward, which falls in contradiction with the change in their interpretation. Moreover, in their figure 3, they clearly show that the rates were higher at 30°S and 40°S during the earliest Miocene, which makes their claim unsustainable.

Note that we are particularly referring to the Late Oligocene (>23 Ma) when trench-normal convergence rates were actually higher at 20°S than further south as seen in Fig. 3. We make this point explicit in the new version of the manuscript (line 192 Changes_Noted) in order to avoid confusions.

On the other hand, I am happy to see that the authors have acknowledged slab anchoring at 660km as a potential mechanism behind the switch from extension to compression throughout the Andes. I also appreciate that the authors consider the motion of the upper-plate as dominant at a larger time-scale, whereas pulses of

deformation can be correlated with changes in convergence related to the motion of the oceanic plate within a continuous contractional regime, such as the one the Andes have been experiencing since Cretaceous times.

Regarding the Andean tectono-magmatic evolution, I must admit that some of my criticisms to the previous version were exaggerated, although my intention was to show that the interpretation of the entire Andes as being built through three synchronous and perfectly timed deformational phases was incorrect. The modification of this interpretation and the clarification that no causal relation is claimed between magmatism and convergence rates has reassured me. Although the magmatic loci and the variations in convergence rates show some degree of correlation, I consider that the geodynamic explanation behind this fact deserves further exploration and debate. Updated shortening rates at 20°S and 30°S adjust well with the new results and, despite the unfortunate non-existence of published rates at 40°S, I believe that the information compiled for Patagonia points towards a similar direction. Regarding the paleoelevation of the Puna plateau, I agree with the authors that the available datasets suggest an episodic uplift, although this debate is far from being closed.

In summary, I consider that the present manuscript constitutes a fine contribution to Nature Communications, and that the high-resolution convergence rates will be useful to address both present and future tectonic and geodynamic debates. I must also admit that the mechanism proposed for convergence variations and tectono-magmatic changes in the upper plate is intriguing, now even more than in the previous version. Therefore, I look forward to seeing the next steps of this project and the further exploration of the delamination hypothesis.

Sincerely,
Dr. Lucas Fennell

We would like to acknowledge again the reviewer for the throughout revision of our work that was so important in order to refine the conceptual geological framework into which to interpret our results, and also for the positive comments to this second version of the manuscript.

Reviewer #3 (Remarks to the Author):

The revised manuscript answers almost all of the reviewer concerns and in my opinion is ready for publication. I found a few minor wording issues in the abstract and text that the authors may wish to change, but all are minor and at the authors' discretion (referenced by Line #):

23 'emerging from' could be "as observed in new"

28 'to revise' could be "revision of"

30 'to' could be "with"

35 'result of hundred' could be "result of one hundred"

63 'hampered the possibility to establish' could be "hampered establishment of"

75 'allow' should be "allows"

92 'benefits' should be "benefit"

(92 - a sentence describing the utility of Redback might be useful here)

98 "of points on the Nazca plate"

116 'very' should be "more"

121 'does occur' could be "occurs"

210 'to' should be "with"

213 'for' should be "to"

214 While "fusing is formally correct here, applying it to the lower and middle crust will imply to some readers that you are proposing to join together the lower and middle crust. Using the verb "melting" would avoid this ambiguity.

217 'drops should be "drop"; also I would omit "down"

222 'with' could be "to"

Thank you very much for your positive comments and to notice the typos that still subsisted in the previous version. We took most of them and incorporated these minor changes in the new version.

REVIEWERS' COMMENTS

Reviewer #1 (Remarks to the Author):

Dear authors,

thankyou again for responding constructively and effectively to the concerns I listed in the second round of review. After reading the responses and the updated version of the manuscript, I am satisfied that it is possible to view the model changes that you present as supported by observations and interpretable in terms of plate driving forces. With this, I have nothing further to add except to wish you every success with this work in the future.

Kind regards,

Graeme Eagles

Response to reviewer comment

Reviewer #1 (Remarks to the Author):

Dear authors,

thankyou again for responding constructively and effectively to the concerns I listed in the second round of review. After reading the responses and the updated version of the manuscript, I am satisfied that it is possible to view the model changes that you present as supported by observations and interpretable in terms of plate driving forces. With this, I have nothing further to add except to wish you every success with this work in the future.

Kind regards,

Graeme Eagles

We are largely grateful to the dedicated and very professional review done by Dr. Eagles to the different versions of our manuscript. His criticism regarding the presentation and interpretation of results, and suggestions for testing the validity of our results were fundamental in order to improve the quality and clarity of our paper. Again, thank you very much for your fruitful collaboration.